# Locally organised and activated Fth1[hi] neutrophils aggravate inflammation of acute lung injury in an IL-10-dependent manner

Kun Wang [1,2], Muyun Wang[1,2], Ximing Liao[1], Shaoyong Gao[1], Jing Hua[1], Xiaodong Wu[1], Qian Guo[1], Wujian Xu[1], Jiaxing Sun[1], Yanan He[1], Qiang Li [1] ✉ & Wei Gao [1] ✉

Acute respiratory distress syndrome (ARDS) is a common respiratory critical syndrome with no effective therapeutic intervention. Neutrophils function in the overwhelming inflammatory process of acute lung injury (ALI) caused by ARDS; however, the phenotypic heterogeneity of pulmonary neutrophils in ALI/ARDS remains largely unknown. Here, using single-cell RNA sequencing, we identify two transcriptionally and functionally heterogeneous neutrophil populations (Fth1[hi] Neu and Prok2[hi] Neu) with distinct locations in LPS-induced ALI mouse lungs. Exposure to LPS promotes the Fth1[hi] Neu subtype, with more inflammatory factors, stronger antioxidant, and decreased apoptosis under the regulation of interleukin-10. Furthermore, prolonged retention of Fth1[hi] Neu within lung tissue aggravates inflammatory injury throughout the development of ALI/ARDS. Notably, ARDS patients have high ratios of Fth1 to Prok2 expression in pulmonary neutrophils, suggesting that the Fth1[hi] Neu population may promote the pathological development and provide a marker of poor outcome.

Acute respiratory distress syndrome (ARDS) is a severe form of acute lung injury (ALI) characterized by endothelia-epithelial barrier disruption, alveolar damage and pulmonary edema, often leading to hypoxemic respiratory failure[1]. Despite advances in critical care and organ supportive techniques, ARDS remains a prominent socioeconomic burden, with no effective therapeutic intervention and a hospital mortality rate of greater than 40%[2].

Neutrophils have long been recognized the core immune effector cells in the pathogenesis, progression and resolution of numerous diseases, including ALI/ARDS[3]. As a driving force in the inflammatory process, neutrophils represent the first line of defense in innate immune responses, producing antibacterial peptides, reactive oxygen species, cytokines and other inflammatory mediators; nevertheless, exaggerated neutrophil accumulation and prolonged activation of neutrophils can have deleterious effects on tissues, thus exacerbating ALI/ARDS[4]. In contrast to the other components of the innate immune system that are represented by macrophages and dendritic cells, neutrophils are traditionally thought to constitute a homogenous

population with well-defined and highly conserved function. However, increasing evidence over the last decade has demonstrated unexpected phenotypic heterogeneity and functional versatility among neutrophil populations[5]. For example, mature neutrophils (CD16[high]CD62L[high]), immature neutrophils (CD16[low]CD62L[high]) and suppressive neutrophils (CD16[high]CD62L[low]) have been identified in the circulation of humans with trauma or systemic lipopolysaccharide (LPS) challenge[6]. A subset of Ly6G[+] neutrophils expressing high level of CD11b has been reported to retain within the pulmonary microvascular system under endotoxin infection and be capable of capturing disseminating pathogens[7]. During the global pandemic of coronavirus disease 2019 (COVID-19), researchers identified the emergence of a low-density neutrophil population expressing intermediate levels of CD16 from the peripheral blood and bronchial lavage fluid (BALF) of COVID-19-associated ARDS patients[8,9]. Such type of neutrophils demonstrated enhanced phagocytosis, neutrophil extracellular trap (NET) formation, platelet activation, cytokine production, T-cell activation, and impaired degranulation[10,11], thus were considered as a predictor of risk for worse

---

[1]Department of Pulmonary and Critical Care Medicine, Shanghai East Hospital, Tongji University School of Medicine, Shanghai 200120, China. [2]These authors contributed equally: Kun Wang, Muyun Wang. ✉e-mail: liqressh1962@163.com; grace19881118@126.com

outcomes for COVID-19 patients. Furthermore, alveolar and circulating neutrophils in ARDS patients show delayed apoptosis as well as dramatic transcriptional profile alterations that confer resistance to phosphoinositide 3-kinase inhibition[12]. As a canonical immunosuppressive factor, interleukin-10 (IL-10) has been shown to relieve excess inflammation and reduce tissue damage in various disorders[13]. Additionally, IL-10 inhibits the chemotaxis and aggregation of neutrophils, reducing their pro-inflammatory cytokine production and suppressing LPS-increased cell vitality[14]. However, distinct subsets for the most abundant leukocytes within the lungs have yet to be classified during ALI/ARDS, and the mechanisms by which IL-10 modulates neutrophils in ALI/ARDS remain largely unexplored.

During ARDS, the transcriptional signatures and phenotypic properties of neutrophils, which are reshaped by inflammatory stimuli, are thought to be impacted by the migration process from the bone marrow via circulation to inflamed sites. Insulting factors such as LPS induce neutrophil entrapment in pulmonary capillaries, followed by activation, sequestration into interstitium, and trans-epithelial migration. Unlike the classic pathway of neutrophil recruitment, entrapment and migration of these cells in the lungs are somewhat unique[15]. The morphological structure and size (diameter from 2 to 15 μm) of the pulmonary capillary network necessitate those neutrophils (ranging from 6 to 10 μm) deform significantly in order to pass through the tighter (5 μm) segments[16]. Furthermore, mechanical deformation can rapidly alter the activation state of neutrophils[17]. Therefore, it is likely that the transcriptional profile and functional characteristics of recruited neutrophils vary according to their intra-pulmonary distribution, though potential differences in neutrophil subsets during ALI/ARDS have not been well defined.

Ferritin heavy chain (Fth1 or FHC), encoded by the *Fth1* gene, is a cytosolic iron storage protein regulating multiple physiological process including ferroptosis, autophagy, oxidative stress and inflammation[18,19]. Iron is toxic due to its generation of reactive species that can directly damage DNA and proteins[20], and ferritin captures and buffers the intracellular labile iron pools (LIP), thus preventing cells from oxidative injury[21,22]. Increased Fth1 expression in cardiomyocytes[23] or embryonic fibroblasts[24] has been proved to help maintain the iron balance and mitochondrial homeostasis, thereby protecting cells against oxidative damage. Prok2 (Prokineticin 2), also known as the ortholog of Bv8, is highly expressed in neutrophils from peripheral blood and bone marrow, which in turn regulates neutrophil migration[25]. Prok2 is also overexpressed in infiltrating neutrophils of inflamed tissues, thus providing evidence linking expression and function of Prok2 to the innate immune system[26]. Up to now, the involvement of Fth1 and Prok2 in the pathogenesis of ALI/ARDS has not been studied in depth.

In this work, we apply single-cell RNA sequencing (scRNA-seq) technology to classify transcriptional variation among neutrophils in an ALI mouse model for ARDS. Our findings indicate that pulmonary neutrophils can be subdivided into two major populations based on their differential gene expression patterns (Fth1hi Neu and Prok2hi Neu). These two cell populations are located at distinct compartments of the lung, with differences in glyco-metabolism, antimicrobial peptide and inflammatory factor production, antioxidant ability, and programmed cell death. We further demonstrate that the quantity, function and fate of the distinct neutrophil subsets are regulated by IL-10. Finally, we add clinical evidence for the relevance of these findings in ARDS patients.

## Results

### IL-10 deficiency aggravates acute neutrophil inflammation in the ALI mouse model

To explore the activity of IL-10 in neutrophil-mediated inflammatory processes in ARDS, we employed an ALI mouse model induced by intranasal LPS administration (10 mg/kg). Wild type (WT) and *Il-10*−/− mice were treated with LPS or PBS, followed by analyses at 6 h, 12 h and

24 h after administration (Fig. 1a). As shown in Fig. 1b–e, LPS induced protein accumulation and inflammatory cell (especially neutrophil) infiltration in BALF, which was further exacerbated by IL-10 depletion after 12 h. To elucidate the local and systematic trend of inflammation, we measured levels of pro-inflammatory cytokines in BALF supernatant and serum. Interleukin-6 (IL-6), keratinocyte chemoattractant (KC), tumor necrosis factor (TNF) and granulocyte colony-stimulating factor (G-CSF) were elevated rapidly within 6–12 h (Fig. 1f–i), while IL-12p70, interferon-γ (IFN-γ), IL-1β and IL-17A reached a peak at 24 h (Fig. 1j–m). In most cases, IL-10 deficiency intensified the elevation of these inflammatory mediators at peak time points. Meanwhile, we estimated IL-10 levels in WT mice and found that IL-10 content was low in BALF supernatant, which further reduced 24 h after LPS administration (Supplementary Fig. 1a). As for serum IL-10, it elevated at 6 h after LPS challenge, followed by a decline in 12 and 24 h (Supplementary Fig. 1b). Interestingly, it was accompanied by the changes of IL-6, KC and TNF, which reached a peak at 6 h in WT-LPS mice and further increased in *Il-10*−/−-LPS mice, suggesting a protective function for IL-10 in the acute inflammation of ALI.

To further examine the effect of IL-10 on the cellular milieu of the lungs during LPS-induced ALI, we evaluated the histology of the injured lungs according to 5 major dimensions: alveolar neutrophils, interstitial neutrophils, hyaline membranes, proteinaceous debris filling the airspaces, and alveolar septal thickening[27]. The results indicated that IL-10 deficiency did not initiate pulmonary inflammation under physiological condition; however, extensively exacerbates progressive lung injury after 24 h induction with LPS (Fig. 2a, b), which can also be observed as early as 12 h after LPS induction (Supplementary Fig. 2). The permeability of alveolocapillary membrane in the ALI mice was then estimated by calculating the ratio of wet lung to dry lung (W/D ratio). It was found that lack of IL-10 enhanced W/D ratio of lung tissues (Fig. 2c).

To evaluate the IL-10-dependent accumulation and activation of neutrophils upon LPS-mediated lung injury, we stained mice lung sections with Ly6G, myeloperoxidase (MPO) and neutrophil elastase (NE), each of which are key constituents of azurophilic granules, a primary secretory organelle regulating adhesion, phagocytosis and NET formation[28]. After LPS challenge, infiltrated neutrophils expressed higher levels of all three proteins in *Il-10*−/− mice compared with WT mice (Fig. 2d–i). Besides, the effect of IL-10 depletion on survival rate of ALI mice was also evaluated with lethal doses of LPS (25 mg/kg). As shown in Supplementary Fig. 3, *Il-10*−/− mice had a lower survival rate (26.7%) than WT mice (66.7%), indicating a potential impact of IL-10 level on ALI/ARDS prognosis.

To evaluate the prophylactic potential of IL-10 for ALI, WT mice were co-injected nasally with LPS in combination with recombinant IL-10 (rIL-10) or PBS based on the previous literature[29]. The mice of receiving LPS + rIL-10 administration had less neutrophil-predominant inflammation and reduced IL-6 levels (Supplementary Fig. 4). Furthermore, rIL-10 treatment decreased the diffuse damage in ALI lungs and the numbers of Ly6G+, MPO+ and NE+ neutrophils (Supplementary Fig. 5). These data verify that neutrophil chemotaxis and activation within ALI lungs are susceptible to IL-10.

### Transcriptionally heterogeneous neutrophil populations in WT and *Il-10*−/− ALI mouse lung show distinct patterns of tissue distribution

To obtain a high-resolution map of the mouse lung under normal and pathological conditions, we employed the scRNA-seq method. Lung tissues of WT and *Il-10*−/− mice were obtained 24 h after LPS or PBS inhalation, rapidly digested to a single-cell suspension and analyzed following a single-tube protocol with unique transcript counting through barcoding with unique molecular identifiers (UMIs) using 10× Genomics Chromium platform (Fig. 3a). After quality filtering, a total of 18,146 cells were analyzed for transcriptional profiling, for which an

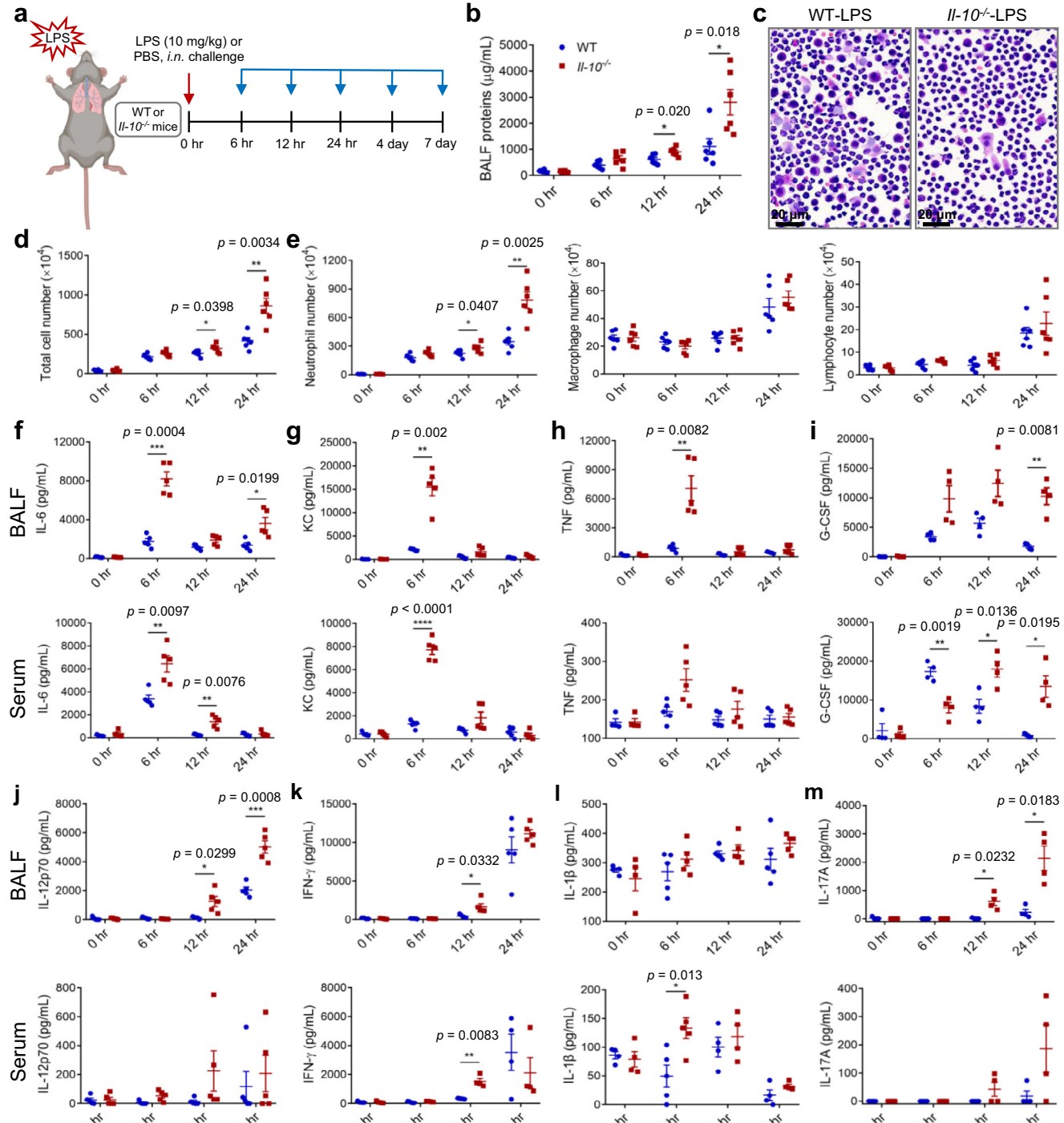

**Fig. 1 | IL-10 depletion aggravates neutrophil-predominant inflammation following LPS challenge. a** Experimental strategy for morphological and functional analysis of neutrophils after ALI. WT and *Il-10⁻/⁻* mice were treated nasally with 10 mg/kg LPS or PBS, followed by analyses at 6 h, 12 h, 24 h, 4 d and 7 d after administration. Some figure elements were created with BioRender.com. **b–e** Animals were euthanized and BALF was evaluated for total protein (**b**), total cell number (**d**), neutrophil, macrophage and lymphocyte numbers (**e**) at 6 h, 12 h, 24 h, n = 6 in each group; as well as H&E cell staining (**d**) at 24 h after LPS challenge. Scale bars, 20 µm. **f–m** Production of cytokines in BALF and serum was evaluated for IL-6 (**f**, n = 4, 5, 5 in each time point from WT and *Il-10⁻/⁻* group both in BALF and serum), KC (**g**, n = 4, 5, 5, 5 in each time point from WT and *Il-10⁻/⁻* group both in BALF and serum), TNF (**h**, n = 4, 5, 5, 5 in each time point from WT and *Il-10⁻/⁻* group both in BALF and serum), G-CSF (**i**, n = 4 in each group both in BALF and serum), IL-12p70 (**j**, n = 5 in each group both in BALF and serum), IFN-γ (**k**, n = 4, 5, 5, 5 in each time point from WT and *Il-10⁻/⁻* group in BALF; n = 4, 5, 4, 4 in each time point from WT and Il-10⁻/⁻ group in serum), IL-1β (**l**, n = 4, 5, 5, 5 in each time point from WT and *Il-10⁻/⁻* group in BALF; n = 4, 5, 4, 4 in each time point from WT and *Il-10⁻/⁻* group in serum) and IL-17A (**m**, n = 4 in each group both in BALF and serum). All samples were biologically independent and three or more independent experiments with similar results were performed. Data are presented as mean ± SEM and analyzed with a 95% confidence interval. Statistical analysis was performed using two-tailed unpaired Student t test. Source data are provided as a Source Data file.

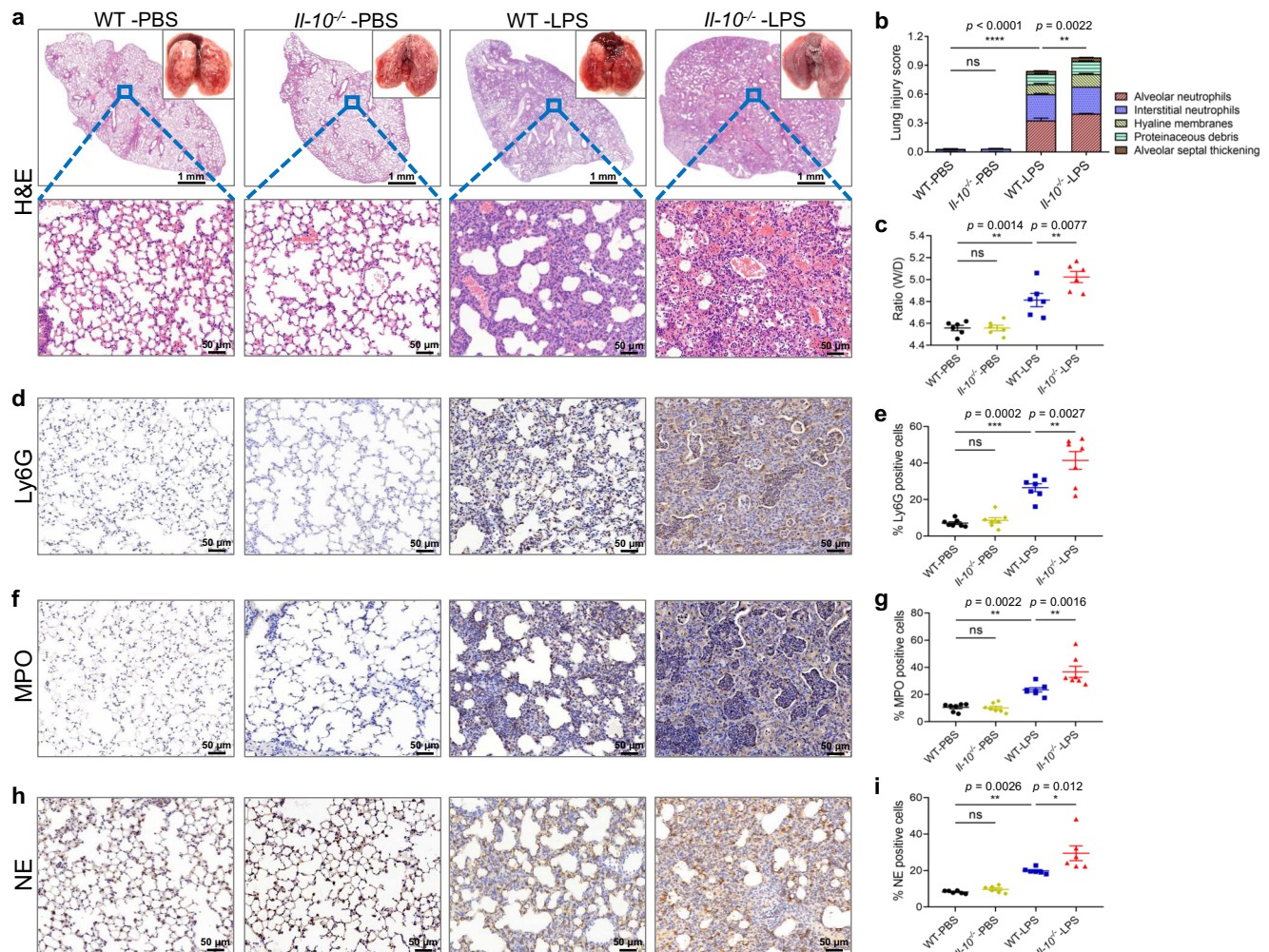

**Fig. 2 | IL-10 deficiency exacerbates the extensive lung injury induced by LPS.**
**a** Representative images of lungs and H&E-stained tissue sections at 24 h after LPS stimulation. Scale bars in the upper panel, 1 mm; scale bars in the below panel, 50 μm. **b** Lung damage was determined by 5 pathophysiological features to obtain the total injury score, $n = 6$ in each group. **c** The permeability of alveolocapillary membrane was evaluated by the lung wet/dry (W/D) ratio, $n = 6$ in each group. **d–i** Immunohistochemical staining and quantitative analysis of Ly6G (**d**, **e**, $n = 7$ in each group), MPO (**f–g**, $n = 7$ in each group) and NE (**h**, **i**, $n = 6$ in each group) in mouse lung samples. ns, not significant. All samples were biologically independent and three or more independent experiments with similar results were performed. Data are presented as mean ± SEM and analyzed with a 95% confidence interval. Statistical analysis was performed using two-way ANOVA followed by Bonferroni's post hoc test. Source data are provided as a Source Data file.

average of 1,268 genes per cell were measured. Among them, 1,129 cells from WT control mice, 4,478 cells from WT LPS-treated mice, and 6,518 cells from $Il$-$10^{-/-}$ LPS-treated mice were profiled (Fig. 3b). By using nonlinear dimensional reduction and graph-based clustering strategies, we visualized 18 subsets in two-dimensional t-distributed stochastic neighborhood embedding (t-SNE) projections (Fig. 3c). We next sought out prototypical markers that defined cell populations and categorized these subsets into 9 major clusters, comprising epithelial cells ($Epcam$), endothelial cells ($Pecam1$), fibroblasts ($Col1a2$), neutrophils ($Ly6g$), monocyte-macrophages ($Fcgr1$, $Mrc1$), lymphocyte T-cells ($Cd3e$), lymphocyte B-cells ($Cd19$), dendritic cells ($Cd83$) and natural killer cells ($Klrb1c$) (Fig. 3d–f; Supplementary Fig. 6). These cell types demonstrated distinct top 15 enriched genes with typical corresponding biological functions (Fig. 3g). Notably, we observed increased proportion of neutrophils after LPS stimulation, and discrepant neutrophil subtypes could be identified between WT and $Il$-$10^{-/-}$ ALI mice lungs.

Since neutrophils represented the largest cell cluster in ALI lungs (Fig. 3d), we sought to explore the transcriptional properties of neutrophils that are altered during disease pathogenesis. We identified

7 transcriptionally distinct clusters of neutrophils, including N1 (35.5%, highly expressed $Fth1$ and $S100a8$), N2 (34.4%, highly expressed $Fth1$ and $S100a8$), N5 (17.9%, highly expressed $Fth1$ and $S100a8$), N6 (12.2%, highly expressed $S100a8$ and $S100a9$) from $Il$-$10^{-/-}$ ALI mice; and N3 (42.3%, highly expressed $Fth1$ and $S100a8$), N4 (37.8%, highly expressed $Fth1$ and $S100a8$), N7 (19.9%, highly expressed $S100a8$ and $S100a9$) from WT ALI mice (Fig. 4a, b; Supplementary Fig. 7). To characterize the phenotypes of these subsets in detail, we performed single-cell differential expression analysis (SCDE) for each population and detected characteristic gene expression patterns for N1-N5 and N6-N7 (Fig. 4c; Supplementary Fig. 8a, b). Based on their specifically enriched genes, we classified the neutrophils from ALI lungs into two populations: N1, N2, N5 ($Il$-$10^{-/-}$) and N3, N4 (WT) highly expressed $Fth1$ and other genes ($Il12a$, $Adora2b$, etc) and were therefore referred to as Fth1$^{hi}$ Neu; N6 ($Il$-$10^{-/-}$) and N7 (WT) highly expressed $Prok2$, as well as $Scrg1$ and $Serpinb1a$, and were defined as Prok2$^{hi}$ Neu (Fig. 4d–g).

Next, we sought to investigate whether the enriched genes could be used to distinguish the two cell populations in ALI lungs according to their relative tissue distribution. Using immunohistochemistry and immunofluorescence assays, we determined that Fth1 could be

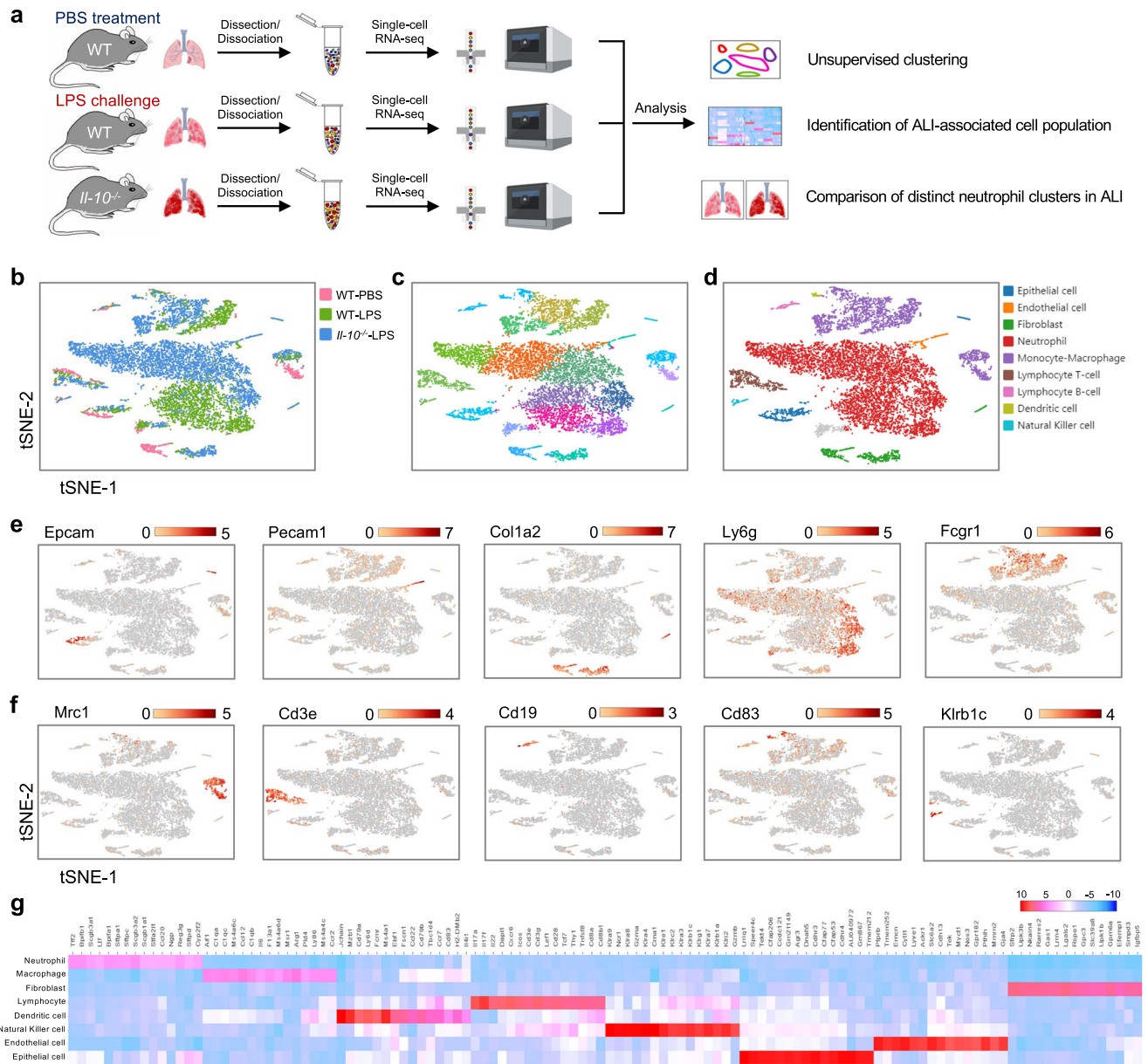

**Fig. 3 | Identification of distinct cell populations in mouse lung tissues by Single-cell RNA sequencing. a** Schematic diagram of our experimental design. Some figure elements were created with BioRender.com. **b, c** The t-SNE representation of aligned gene expression data in single cells collected from WT control (1,129 cells), WT LPS-treated (4,478 cells) and $Il$-$10^{-/-}$ LPS-treated (6,518 cells) mouse lung samples showed cellular origins (**b**) and partitioned into 18 distinct clusters (**c**). **d** Unsupervised clustering identified 9 major pulmonary cell subsets. Each point depicts a single cell, colored based on the cluster designation. **e, f** Gene expression profiles projected onto t-SNE plots for *Epcam, Pecam1, Col1a2, Ly6g, Fcgr1, Mrc1, Cd3e, Cd19, Cd83* and *Klrb1c* (scale: log-transformed gene expression). **g** A heatmap exhibiting the 15 most enriched genes used for biological identification of each cluster defined in panel D (scale: Log2 fold change). WT-PBS, $n = 1$ sample; WT-LPS, $n = 2$ samples; KO-LPS, $n = 2$ samples.

abundantly detected in neutrophils within the bronchi, alveolar space and septum, but not those of vessels (Fig. 4h–j, n; Supplementary Fig. 8c–e). Instead, Prok2 was preferentially expressed in neutrophils from blood vessels (Fig. 4k–n). Taken together, the ALI lungs appear to be comprised of two predominate neutrophil populations: Prok2^hi Neu (a minority population that localizes within pulmonary vessels) and Fth1^hi Neu (a majority population distributed widely in or around airways).

### Prok2^hi Neu and Fth1^hi Neu subsets in ALI mice lungs are functionally and morphologically heterogeneous

To characterize the functional heterogeneity of the two neutrophil populations found in ALI mice lungs, we performed Gene Ontology (GO) enrichment and Kyoto Encyclopedia of Genes and Genomes (KEGG) pathway analyses (Supplementary Fig. 9). Most of the biological processes were overlapping in the neutrophil subsets, as expected by their similar lineages. Interestingly, however, Prok2^hi Neu were enriched in several specific functional pathways that were barely detectable in Fth1^hi Neu, including apoptosis and ferroptosis pathways (Supplementary Fig. 9, red boxes), suggesting that retention in lung tissues may protect Fth1^hi Neu from programmed cell death. Consistently, Fth1^hi Neu exhibited lower expression of L-selectin (*CD62L*) and C-X-C motif receptor 2 (*Cxcr2*), and higher expression of *Cxcr4* and intercellular adhesion molecule-1 (*Icam1*) (Fig. 5a, b), indicating a predominate 'senescence' phenotype[30]. Morphologically, Prok2^hi Neu displayed a ring-form nucleus, which is closer to that of blood

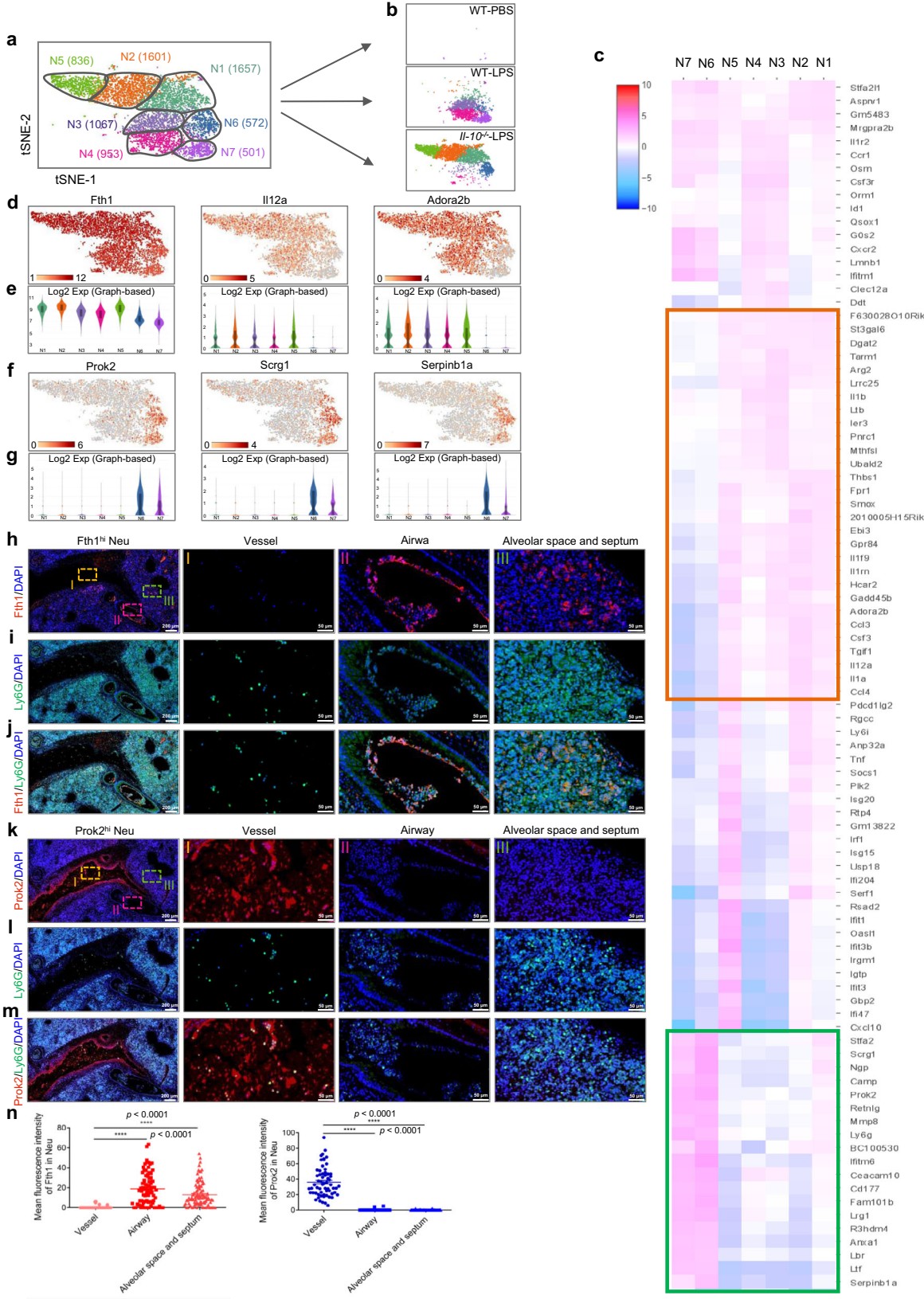

neutrophils, while Fth1hi Neu tended to have a multi-lobular or even hyper-segmented nucleus (Fig. 5c).

To further explore functional differences between the two phenotypes, we standardized the gene expression according to the Z-scores. The Fth1hi cells exhibited higher expression in genes primarily associated with chemotaxis (*Ccl3, Ccl4, Fpr1*), pro-inflammatory cytokines (*Il1a, Il12a, Tnf*), inhibitors of apoptosis (*Bcl2a1a, Bcl2a1b, Bcl2a1d*), maintenance of iron homeostasis (*Fth1, Ftl1, Slc7a11, Slc3a2*) and anti-oxidative stress (*Sod2, Hmox1, Gstm1*)[31]. Comparatively, Prok2hi Neu displayed a gene expression signature suggestive of anti-bacterial innate immunity (*Camp, Ltf*), defense response to pathogens (*Ngp, Padi4, Serpinb1a, Prok2, S100a8, S100a9*), clearance of apoptotic

**Fig. 4 | Transcriptional and distribution heterogeneity of neutrophils in lungs of ALI mice. a, b** Two-dimensional visualization of single-cell clustering of neutrophil profiles inferred from the RNA-seq data for all neutrophils in normal and ALI lung samples. **c** Heatmap of enriched genes in each neutrophil subtype versus the others using SCDE analysis. The orange box indicates similar genes expressed in N1-N5; the green box indicates similar genes expressed in N6-N7. **d–g** Gene expression patterns projected onto t-SNE plots (**d, f**) and corresponding violin plots (**e, g**) for *Fth1, Il12a, Adora2b, Prok2, Scrg1* and *Serpinb1a* (scale: log-transformed gene expression). Boxes within violin plot show the median ± 1 quartile, with the whiskers extending from the hinge to the smallest or largest value within 1.5× interquartile range from the box boundaries. *n* = 1657, 1601, 1067, 953, 836, 572, 501 for N1-N7.

**h–m** Immunofluorescence staining with Fth1/Ly6G (**h–j**) and Prok2/Ly6G (**k–m**) for detecting the distribution of distinct neutrophil subsets in ALI lung samples. Nuclei were stained with DAPI (4′, 6-Diamidino-2-phenylindole), displayed in blue. Scale bars in the left column, 200 μm; scale bars in the right three columns, 50 μm. **n** Quantitative analysis of the mean fluorescence intensity of Fth1 (*n* = 76, 77, 97, respectively from 6 mice) and Prok2 (*n* = 62, 62, 70, respectively from 6 mice) in mouse lung samples. All samples were biologically independent and three or more independent experiments with similar results were performed. Data are presented as mean ± SEM and analyzed with a 95% confidence interval. Statistical analysis was performed using one-way ANOVA followed by Bonferroni's post hoc test. Source data are provided as a Source Data file.

cells (*Anxa1, Itgb2l*) and glyco-metabolism processes (*G6pdx, Hk3, Ldha, Mpc1, Slc2a3*)[32,33] (Fig. 5d). To explore whether Prok2[hi] Neu in the pulmonary vascular system differ from peripheral blood neutrophils, we also compared the expression patterns of the above genes, which demonstrated a similar, but not identical functional phenotype between Prok2[hi] Neu and peripheral blood neutrophils (Fig. 5e).

To further investigate the effects of Fth1/Prok2 on the immune function of neutrophils, we adopted recombinant lentiviral vector of *Fth1/Prok2* shRNA and human promyelocytic leukemia (HL-60) cell-derived neutrophils. After 12 h transfection with lentivirus-mediated shRNA (*Fth1/Prok2*) or negative control shRNA (NEGi), cells were collected to determine their interference efficiency by reverse transcription-polymerase chain reaction and immunoblotting assay (Supplementary Fig. 10a, b). As shown in Supplementary Fig. 10, Fth1-depleted neutrophils exhibited functional defects in anti-oxidation (Supplementary Fig. 10c), anti-apoptosis (Supplementary Fig. 10e) and chemotaxis (Supplementary Fig. 10f), while Prok2 deficient neutrophils defected in reactive oxygen species (ROS) production (Supplementary Fig. 10c), phagocytosis (Supplementary Fig. 10d) and chemotaxis (Supplementary Fig. 10f). Besides, Fth1 deletion inhibited the expression of anti-oxidant HO-1 and the activation of pro-inflammatory NLRP3/Caspase-1 signaling, while elevated pro-apoptotic Bax level compared with the control shRNA, which were not affected by Prok2 (Supplementary Fig. 10g). Overall, this evidence supports the occurrence of two transcriptionally, functionally and morphologically heterogenous neutrophil subsets with distinct tissue distribution within ALI mice lungs.

### Fth1[hi] neutrophils contribute to more severe lung injury in IL-10-depleted mice challenged with LPS

Our results suggest that IL-10 alleviates diffuse damage in ALI lungs by controlling neutrophil-predominant inflammation. Thus, we speculated whether IL-10 may orchestrate differential functions of these two neutrophil clusters after LPS stimulation. Consistently, Fth1[hi] Neu, rather than Prok2[hi] Neu, were found to be elevated in *Il-10*[−/−] compared to WT ALI mice (Fig. 6a, b). Conversely, exogenous IL-10 inhibited Fth1[hi] Neu infiltration into ALI lungs (Supplementary Fig. 11a).

To further validate potential differential responses to IL-10 between these two subsets of neutrophils within ALI mice lungs, we collected BALF neutrophils. IL-10 depletion increased Fth1 levels among airway neutrophils, while co-treatment with rIL-10 reduced their Fth1 expression in LPS-treated mice (Fig. 6d; Supplementary Fig. 11b). On the other hand, Prok2 expression was barely detectable in BALF neutrophils (Fig. 6e), which was consistent with the classification of discrete functional neutrophil populations. Besides, Fth1, rather than Prok2 expression in lung and BALF were positively related to lung injury score (Fig. 6c, f). Thus, Fth1[hi] Neu, represented by airway neutrophils, may be responsible for the aggravated inflammatory damage in ALI lungs that is affected by IL-10.

To evaluate putative underlying mechanisms of IL-10 regulation on Fth1[hi] Neu in ALI, we performed terminal Deoxynucleotide Transferase dUTP Nick End Labeling (TUNEL) staining and flow cytometry, and IL-10 deficiency inhibited the apoptosis of lung Fth1[hi] Neu

(Fig. 6g–h; Supplementary Fig. 11c). Mitochondria has been proved to modulate neutrophil apoptosis via multiple mechanisms, including loss of transmembrane potential, increased membrane permeability and decreased activity of the mitochondrial respiratory chain (MRC) complex[34]. Therefore, we evaluated the mitochondrial morphology of Fth1[hi] Neu by transmission electron microscopy (TEM). At 24 h after LPS exposure, swollen mitochondria with cristae loss could be found in airway neutrophils of WT mice, though similar structural changes were minimal in the *Il-10*[−/−] group (Fig. 6i, j). Furthermore, the activities of MRC complex I and III, which are key redox enzymes responsible for cellular homeostasis, were elevated by IL-10 deficiency in airway neutrophils (Fig. 6k). Therefore, these findings support the possibility that apoptosis resistance in Fth1[hi] Neu is mediated by IL-10 regulation on mitochondrial homeostasis.

Fth1 catalyzes the conversion of ferrous $Fe^{2+}$ into ferric $Fe^{3+}$, thus preventing LIP from participating in oxygen-free radical formation via Fenton chemistry, which is instrumental in avoiding oxidative injury[35]. Accordingly, we speculated that *Il-10*[−/−]-upregulated Fth1 may decrease labile Fe, elevate anti-oxidant capacity and maintain mitochondrial homeostasis, therefore providing an explanation for the reduced apoptosis of Fth1[hi] Neu in ALI lungs. This possibility was confirmed by Prussian blue staining, which exhibited prominent ferric iron deposition in neutrophils of *Il-10*[−/−], but less deposition in WT mice at 24 h postexposure to LPS (Fig. 6l, m). The results indicate that Fth1 may be able to maintain cellular mitochondrial homeostasis of Fth1[hi] Neu in ALI lungs in the absence of IL-10.

### Fth1[hi] neutrophils are essential for unrestrained inflammation in *Il-10*[−/−] mice at later stages of ALI

The inflammatory response and lung injury in WT/*Il-10*[−/−] mice were examined at 4 d and 7 d after LPS exposure, to evaluate the effect of IL-10 and Fth1[hi] neutrophils on the prognosis of ALI. The results demonstrated that the total amounts of inflammatory cells in BALF of *Il-10*[−/−] mice showed a trend toward increase compared to the levels in BALF from WT mice, though the differences were not statistically significant (Supplementary Fig. 12a). However, the levels of IL-6 and IFN-γ were elevated in both BALF and serum; IL-12p70, IL-1β, and IL-17A were elevated in BALF; and G-CSF was elevated in serum after 4 days in *Il-10*[−/−] as compared with WT ALI mice (Fig. 7a–f). As additional pathologic evidence, the lung injury in *Il-10*[−/−] mice was more serious than that in WT mice at both 4 d and 7 d after treatment with LPS (Fig. 7g, h). To investigate whether the poor prognosis in *Il-10*[−/−] mice was associated with neutrophil retention in the lung due to delayed apoptosis of IL-10-induced Fth1[hi] Neu, we performed immunofluorescence assays. The results suggested that abundant Fth1[hi] Neu reside within lung tissue of *Il-10*[−/−] mice as compared to the WT group at 4 d postexposure (Fig. 7i), and this difference was also retained at 7 d postexposure (Fig. 7j). On the other hand, for Prok2[hi] Neu, almost no difference could be observed between the two groups (Fig. 7k, l). Furthermore, Fth1[hi] Neu in the airway and alveolar septum showed decreased apoptosis at day 4 post LPS exposure in *Il-10*[−/−] mice as compared with WT mice, which was not observed at day 7 by TUNEL staining (Supplementary Fig. 12b).

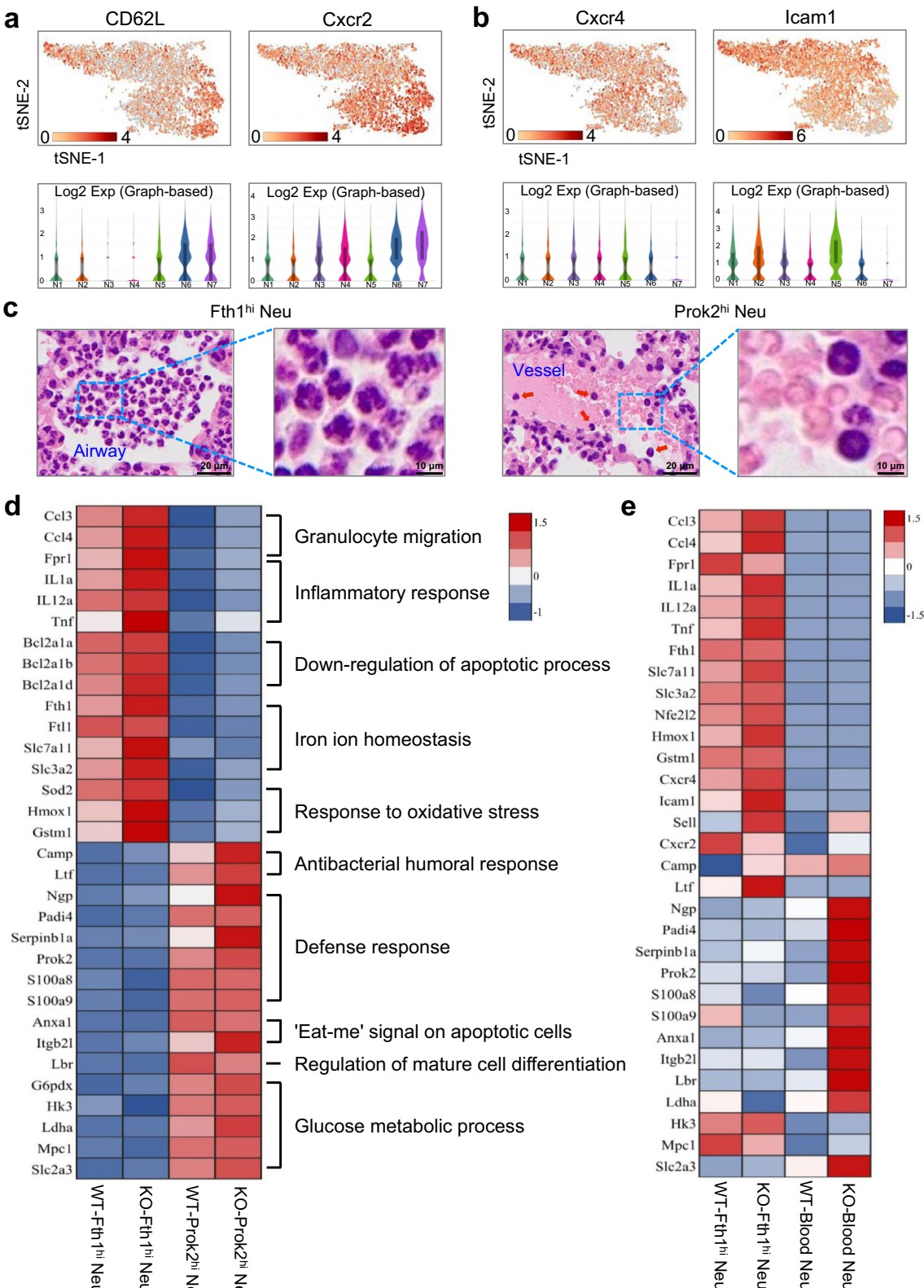

**Fig. 5 | Functional and morphological heterogeneity of two neutrophil subsets in ALI lungs.** **a**, **b** The gene expression profiles projected onto t-SNE plots and corresponding violin plots for *CD62L, Cxcr2, Cxcr4* and *Icam1* (scale: log-transformed gene expression). Boxes within violin plot show the median ± 1 quartile, with the whiskers extending from the hinge to the smallest or largest value within 1.5× interquartile range from the box boundaries. *n* = 1657, 1601, 1067, 953, 836, 572, 501 for N1-N7. **c** Representative micrographs of H&E-stained Fth1hi and

Prok2hi Neu in lung samples from mouse at 24 h post-exposure. Red arrowheads indicate the Prok2hi Neu in pulmonary vessels. Scale bars in the left column, 20 μm; scale bars in the right column, 10 μm. Three independent experiments with similar results were performed. **d**, **e** Heatmaps of selected enriched genes used for biological function identification of each cluster standardized by Z-score. Source data are provided as a Source Data file.

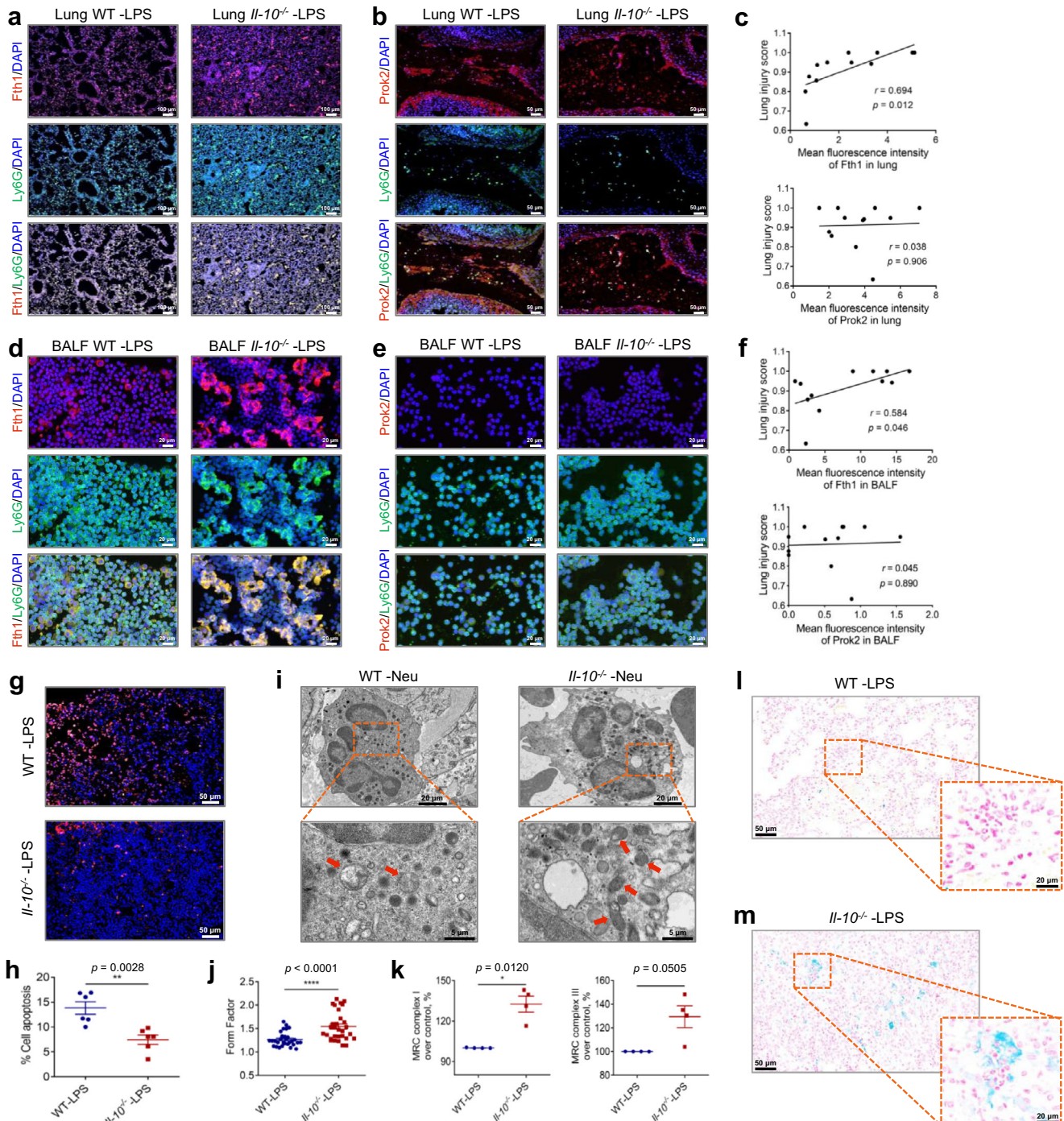

**Fig. 6 | *Il-10⁻/⁻*-prolonged survival of pulmonary Fth1^hi Neu exacerbates lung injury. a, b** Representative images of anti-Fth1/anti-Ly6G (**a**) and anti-Prok2/anti-Ly6G (**b**) immunofluorescence-stained lung tissues of WT and *Il-10⁻/⁻* mice at 24 h after LPS challenge. The nuclei were stained with DAPI, displayed in blue. Scale bars in (**a**), 100 μm; scale bars in (**b**), 50 μm. **c** The relationship between pulmonary Fth1, Prok2 expression and lung injury score, *n* = 12. **d, e** Representative images of anti-Fth1/anti-Ly6G (**d**) and anti-Prok2/anti-Ly6G (**e**) immunofluorescence-stained airway inflammatory cells of WT and *Il-10⁻/⁻* mice at 24 h after LPS stimulation. The nuclei were stained with DAPI, displayed in blue. Scale bars, 20 μm. **f** The relationship between BALF Fth1, Prok2 expression and lung injury score, *n* = 12. **g** Immunofluorescence staining for TUNEL (red) and nuclei (blue) in lung sections at 24 h post-exposure. Scale bars, 50 μm. **h** Quantification of the apoptosis ratio of airway neutrophils from ALI mice by flow cytometry, *n* = 6 in each group.

**i** Representative TEM images showing mitochondrial morphologies of airway neutrophils in WT and *Il-10⁻/⁻* mice 24 h following challenge. Red arrowheads indicate the mitochondria. **j** Form factor analysis of the mitochondria morphological parameter, *n* = 30 in each group. **k** The activity of mitochondrial respiratory chain (MRC) complexes I and III in airway neutrophils of each group, *n* = 4 in each group. **l, m** Microscopic ferric iron deposition in lung sections of WT (**l**) and *Il-10⁻/⁻* (**m**) mice 24 h after LPS administration. All samples were biologically independent and three or more independent experiments with similar results were performed. Data are presented as mean ± SEM and analyzed with a 95% confidence interval. Statistical analysis was performed using two-tailed Pearson's correlation test for (**c, f**), and two-tailed unpaired Student t test for (**h, j, k**). Source data are provided as a Source Data file.

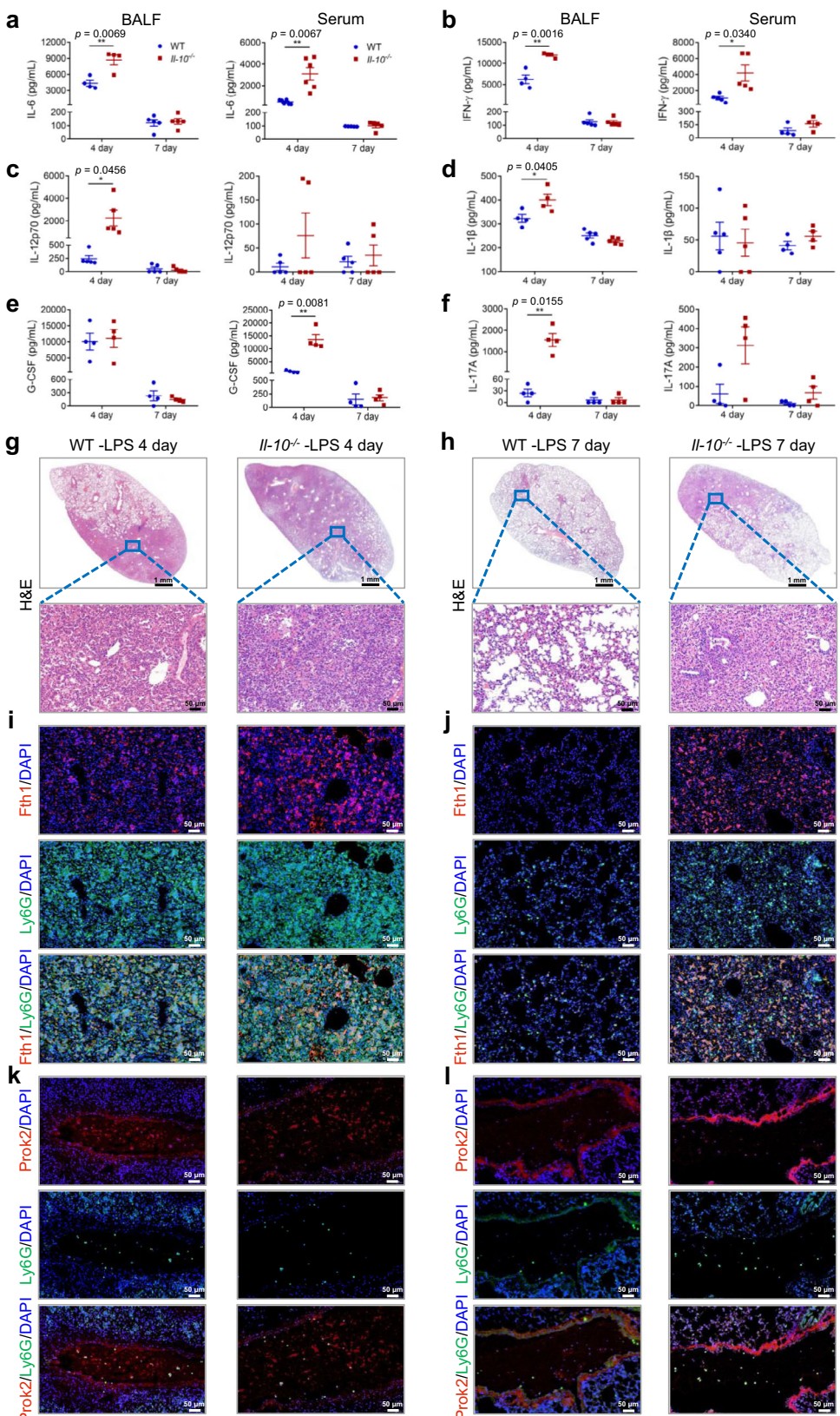

We also used the Smart-Seq method to study global effects of IL-10 on the transcriptome of neutrophils extracted from BALF and blood at various time points after LPS stimulation (Supplementary Fig. 13). The Fth1[hi] Neu had a greater number of up-regulated genes in BALF than in blood neutrophils, while IL-10 depletion further elevated the extent of up-regulation in BALF neutrophils (Fig. 8a–c). To identify biological processes regulated by IL-10 in LPS-treated neutrophils, we performed GO enrichment and KEGG pathway analysis of BALF or blood neutrophils at different times (Supplementary Fig. 14; Fig. 8d–g). The results demonstrated that the Fth1[hi] Neu subset expressed high levels of inflammatory mediators, apoptosis inhibitors and anti-oxidants starting at least at 12 h post LPS challenge (Fig. 8h).

**Fig. 7 | IL-10 depletion leads to poor prognosis of ALI through the persistent harmful lung Fth1[hi] Neu phenotype. a–f** On day 4 and day 7 after LPS challenge, BALF and serum were collected from WT and *Il-10[−/−]* mice for determining the levels of cytokines: IL-6 (**a**, *n* = 4, 4, 5, 5 for BALF and n = 6, 6, 5, 5 for serum), IFN-γ (**b**, *n* = 4, 4, 5, 5 for BALF and *n* = 5, 5, 4, 4 for serum), IL-12p70 (**c**, *n* = 5 in each group for BALF and serum), IL-1β (**d**, *n* = 4, 4, 5, 5 for BALF and *n* = 5, 5, 4, 4 for serum), G-CSF (**e**, *n* = 4 in each group for BALF and serum), and IL-17A (**f**, *n* = 4 in each group for BALF and serum). **g, h** Representative images of H&E-stained tissue sections at 4 d (**g**) and 7 d (**h**) after LPS stimulation. Scale bars in the upper panel, 1 mm; scale bars in the

below panel, 50 μm. **i–l** Representative images of anti-Fth1/anti-Ly6G (**i**, **j**) and anti-Prok2/anti-Ly6G (**k**, **l**) immunofluorescence-stained lung tissues of WT and *Il-10[−/−]* mice at 4 d (**i**, **k**) and 7 d (**j**, **l**) after LPS challenge. The nuclei were stained with DAPI, displayed in blue. Scale bars, 50 μm. All samples were biologically independent and three or more independent experiments with similar results were performed. Data are presented as mean ± SEM and analyzed with a 95% confidence interval. Statistical analysis was performed using two-tailed unpaired Student t test. Source data are provided as a Source Data file.

Comparatively, blood neutrophils exhibited an enhanced expression signature related to antibacterial innate immunity, clearance of apoptotic cells and glucose metabolic processes at 4d and 7d post-exposure in WT mice, though these pathways were activated earlier in *Il-10[−/−]* mice (Fig. 8i). Moreover, the transcription profile of peripheral neutrophils on day 4 was closer to that of Prok2[hi] Neu on day 1, which was consistent with a state of activation of neutrophils upon entrapment from the pulmonary circulation (Fig. 8j–l). Fth1[hi] Neu in BALF as compared to blood neutrophils had sustained high expression of genes related to inflammatory mediators, apoptosis inhibitors and anti-oxidants, which was maintained throughout the time course of ALI development and displayed a further increase in *Il-10[−/−]* mice at day 4 after LPS stimulation. Collectively, these data suggest that the persistent harmful phenotype of lung Fth1[hi] Neu may predict the poor prognosis of ALI, which is modulated by IL-10.

### An increased ratio of Fth1 to Prok2 expression in pulmonary neutrophils is a potential marker for poor outcomes of pulmonary infection

To explore the clinical relevance of our findings, we collected airway neutrophils from BALF of inpatients in our department (n = 24), among which 7 developed into ARDS within 48 h after admission (Supplementary Table 1). For the BALF neutrophils, we observed increased Fth1 and decreased Prok2 expression in the ARDS patients compared to at-risk patients who did not develop the syndrome, including patients with bacterial infections (Fig. 9). These findings support a potential differential role for Fth1[hi] Neu and Prok2[hi] Neu in ARDS and raise the possibility that a high ratio of Fth1 to Prok2 expression in BALF neutrophils from patients with pulmonary infections could provide a marker for short-term poor outcome.

## Discussion

Over the past decade, the importance of neutrophil plasticity has been gradually recognized: immature cells within bone marrow are now thought to migrate into circulation and acquire non-overlapping profiles and functional properties under physiological and pathological conditions, including infection and cancer[36]. Accumulating studies revealed a diverse and complex differentiation landscape of neutrophils within bone marrow and the circulatory system; however, the inter- and intra-tissue heterogeneity of neutrophils remained largely unknown. This gap in understanding could be partly attributed to unique properties of mature neutrophils within tissue, such as limited lifespans and reduced transcriptional activity. Additionally, previous classification approaches have failed to capture the full repertoire of neutrophil subsets that are distinguished by differential compartmentalization and tissue distribution, thus promoting controversy. In this study, we classified neutrophils from ALI lungs into two populations using scRNA-seq: Fth1[hi] Neu (highly expressing *Fth1, Il12a, Adora2b*, etc.) and Prok2[hi] Neu (highly expressing *Prok2, Scrg1, Serpinb1a*, etc.). A major strength is that the differentially enriched genes could be used to distinguish neutrophils subsets with distinct tissue distribution and functional characteristics. Moreover, these two populations were shown to have different functions under pathological conditions. Thus, by considering transcript composition and functional properties together with tissue distribution, we have

provided a more definitive approach to classifying neutrophils within the lungs and have further elucidated processes underlying the development of ALI/ARDS.

In response to infectious or inflammatory signals, neutrophils and other leukocytes are rapidly released from the bone marrow into circulation. Peripheral neutrophils then transmigrate through the endothelium and basal membranes according to a series of processes involving rolling, slowing and adhesion. The transmigrated neutrophils also release mediators that shape the subsequent immune response by modulating adaptive immune cell function[37]. Neutrophils adopt distinct phenotypic and functional properties tailored by the microenvironment and the transmigration process[38]. Recently, Radermecker et al.[39] reported that exposure to a low dose of LPS causes recruited neutrophils to upregulate their *Cxcr4* expression and mediates their release from NETs. These NET-released Cxcr4[hi] neutrophils act as early triggers of type 2 allergic airway inflammation. Our study suggests that when neutrophils migrate from systemic circulation to the pulmonary vasculature in ALI mice, the effect of mechanical deformation and the characteristics of the microenvironment make them more adaptive to defense against pathogens (Prok2[hi] Neu). During subsequent transendothelial and -epithelial movement, large numbers of neutrophils develop into the Fth1[hi] subset, with increased oxidative resistance, delayed apoptosis and greater production of inflammatory mediators (Fig. 10). Interestingly, we observed a more significant difference in the transcriptional and functional profiles between pulmonary intravascular (Prok2[hi] subset) and extravascular (Fth1[hi] subset) neutrophils than between airway and septal Fth1[hi] Neu (Fig. 4; Supplementary Fig. 8). Furthermore, BALF neutrophils isolated from patients with pulmonary infections expressed higher Prok2 and lower Fth1 levels compared to those of ARDS patients (Fig. 9). These data imply that the features of neutrophils might more likely be orchestrated by the local microenvironment rather than by trans-endothelial or -epithelial movements. This possibility will be further verified in future studies.

Our results also suggest that neutrophil heterogeneity within the lungs during ALI development is largely determined by IL-10-mediated signaling. As an immunosuppressive cytokine, IL-10 is mainly produced by activated immune cells and has a critical function in controlling overactivated inflammation[29]. Exogenous recombinant IL-10 (rIL-10) has been demonstrated to effectively ameliorate lung injury in ALI mice[40], which is supported by our results. Neutrophils are major target cells for IL-10 anti-inflammatory effects in the LPS-induced sepsis model[41], which could potentially be explained by accelerated neutrophil apoptosis induced by IL-10 upon LPS stimulation[42], though the relevance of these findings to ALI was unclear prior to this study. We observed decreased Fth1[hi] Neu apoptosis in lungs of *Il-10[−/−]* mice on day 1 and day 4 post-exposure (Fig. 6g, h; Supplementary Fig. 11c; Supplementary Fig. 12b), thus extending the knowledge of the later phase of ALI development. Additionally, our results indicate that this delayed apoptosis is associated with the regulation of iron metabolism, maintenance of mitochondrial homeostasis and stronger antioxidant ability, accompanied by aggravated inflammation in ALI mouse lungs (Fig. 5d, e; Fig. 6i–m). Our study, therefore, deepens our understanding of mechanisms regarding the anti-inflammatory effect of IL-10.

Herein, we identified *Fth1* and *Prok2* as potential gene markers for neutrophil subsets in lung tissues of LPS-stimulated ALI mice using

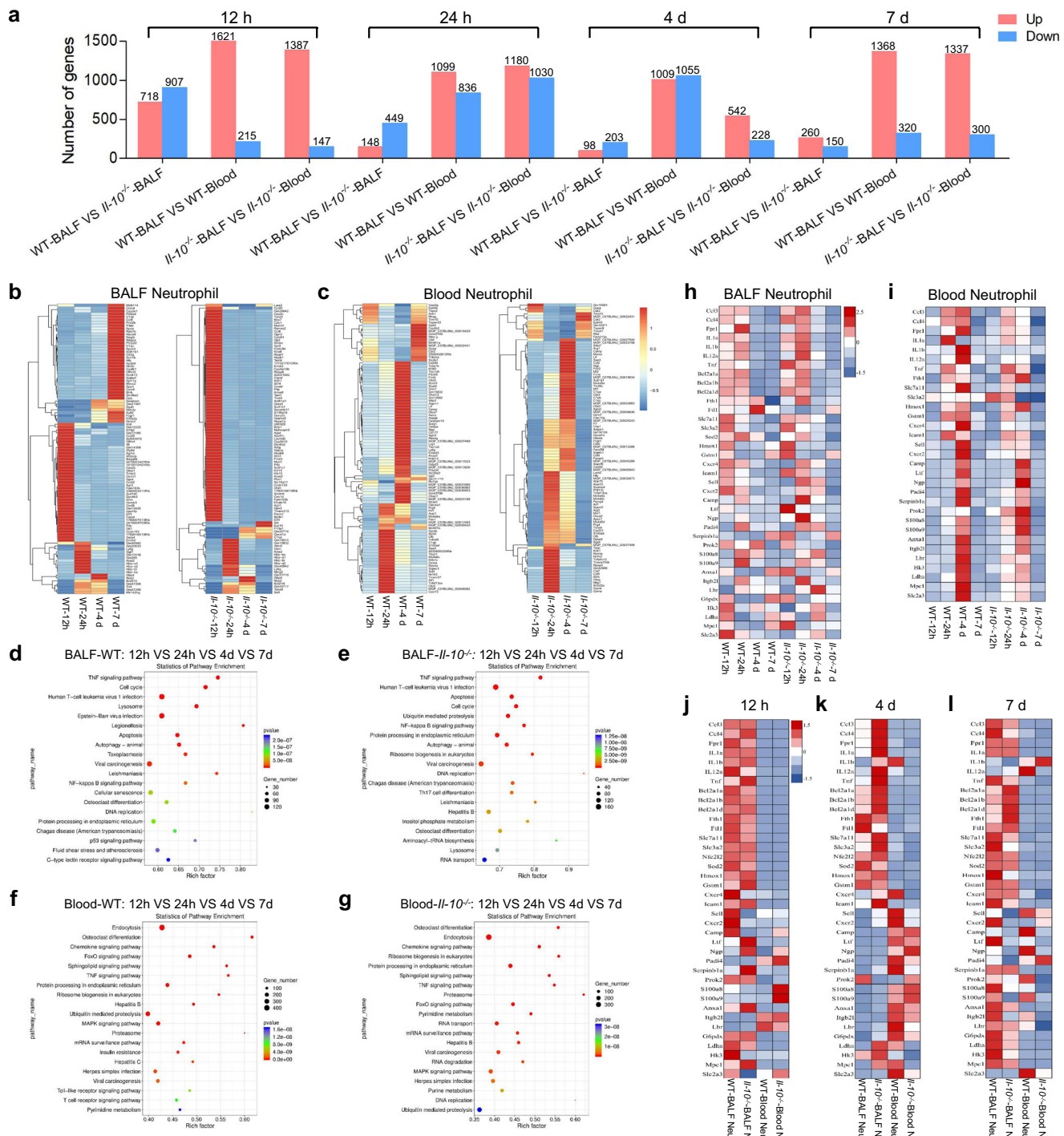

**Fig. 8 | Global effects of IL-10 on the transcriptomes of neutrophils extracted from BALF and blood at various time points after LPS stimulation. a** Numbers of differentially expressed genes in neutrophils from various groups. **b, c** Heatmaps showing the most highly differentially expressed genes in BALF (**b**) and blood (**c**) neutrophils from different groups. **d–g** KEGG pathway analyses of BALF (**d, e**) and blood (**f, g**) neutrophils from different treatment groups. The statistical analysis was performed by Fisher's test. **h, i** Heatmaps of selected enriched genes used for biological identification of BALF (**h**) and blood (**i**) neutrophils after different treatments, standardized by Z-score. **j–l** Heatmaps of selected enriched genes used for biological identification of neutrophils from distinct groups at 12 h (**j**), 4 d (**k**) and 7 d (**l**) post-exposure. *n* = 4 samples per group. VS, versus. Source data are provided as a Source Data file.

single-cell sequencing. We also demonstrated that Fth1$^{hi}$ Neu, rather than Prok2$^{hi}$ Neu, aggravate inflammatory damage throughout the course of development of ALI. In this study, strongly expressed Fth1 in the airway and alveolar septal neutrophils (Fig. 4h–j, n) was demonstrated to diminish labile Fe$^{2+}$, enhance antioxidant capability, and ameliorate mitochondrial dysfunction under LPS stimulation, thus prolonging the life span of the Fth1$^{hi}$ Neu subset. Previously, Takata

and colleagues reported an increase neutrophil subset within the alveolar and interstitial compartments on day 1 during acid-induced lung injury[43], which shared the similar intra-pulmonary location with the Fth1$^{hi}$ neutrophils, though needed to be further validated. Prok2 was mainly observed in neutrophils within pulmonary vessels (Fig. 4k–n), which indicates a potential function for Prok2$^{hi}$ Neu in the defense against pathogens. A previous study demonstrated that

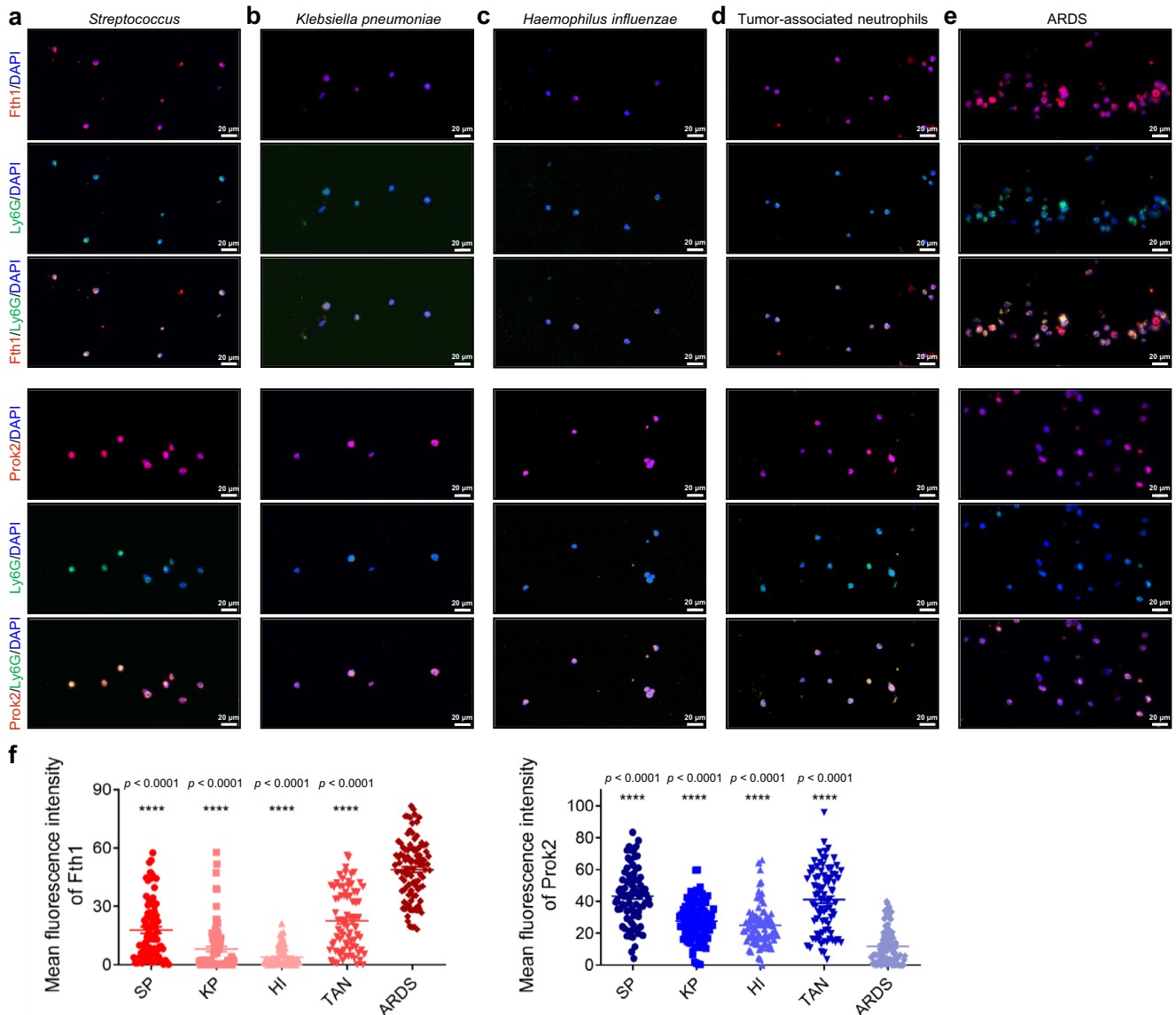

**Fig. 9 | Increased Fth1 and decreased Prok2 expression in BALF neutrophils from patients with pulmonary infections could provide a potential marker for poor outcome. a–e** Representative fluorescence images of anti-Fth1/anti-Ly6G and anti-Prok2/anti-Ly6G-stained airway neutrophils from patients infected with *Streptococcus pneumoniae* (**a**), *Klebsiella pneumoniae* (**b**) and *Haemophilus influenzae* (**c**), as well as patients suffering from pulmonary tumors (**d**) and ARDS (**e**). The nuclei were stained with DAPI, displayed in blue. Scale bars, 20 μm. (**f**) Quantification of the fluorescence intensity in each neutrophil showing the level of Fth1 (*n* = 81 from 3 patients, 81 from 4 patients, 81 from 3 patients, 81 from 3 patients, 100 from 7 patients, respectively in each group) and Prok2 (*n* = 84 from 3 patients, 90 from 4 patients, 83 from 3 patients, 83 from 3 patients, 101 from 7 patients, respectively in each group). SP, *Streptococcus pneumoniae*. KP, *Klebsiella pneumoniae*. HI, *Haemophilus influenzae*. TAN, tumor-associated neutrophils. All the samples were biologically independent and three or more independent experiments with similar results were performed. Data are presented as mean ± SEM and analyzed with a 95% confidence interval. Statistical analysis was performed using one-way ANOVA followed by Bonferroni's post hoc test. Source data are provided as a Source Data file.

vascular neutrophils were rapidly arrested in the pulmonary micro-vasculature after bacterial stimulation[44]. Kubes and colleagues reported that pulmonary neutrophils sequestered within the capillaries after intravenous endotoxin challenge in a CD11b-dependent process and eliminated bloodstream pathogens[7]. Although the transcriptome comparison is absent, Prok2hi Neu identified in the present study seems to be closely related to capillaries-sequestered neutrophils defined by Kubes group on account of similar location and functional characteristics. We carried out a primary attempt to explore the potential regulation effect of Fth1/Prok2 on pulmonary neutrophil functional and phenotypical regulation during ALI development and we are now constructing *Fth1*-eCKO1 mice to further validate our theories.

We anticipate that increased Fth1 and decreased Prok2 expression in patients' BALF neutrophils could provide a potential marker for worse outcomes of pulmonary infection, though additional studies with larger sample size will be needed to confirm this possibility. Elevated serum ferritin has been associated with coronary artery disease, sideroblastic anemia, malignancy, hemophagocytic syndrome and poor outcomes following stem cell transplantation[45,46]. Connelly and colleagues[47] reported that serum ferritin levels are elevated in patients at risk for the development of ARDS; and Zhou et al.[48] demonstrated that elevated serum ferritin is a potential risk factor for COVID-19 mortality. However, the use of serum ferritin as a circulating biomarker for ARDS has not yet been widely accepted in clinical practice. A major limitation could be that the levels of serum ferritin are affected under a

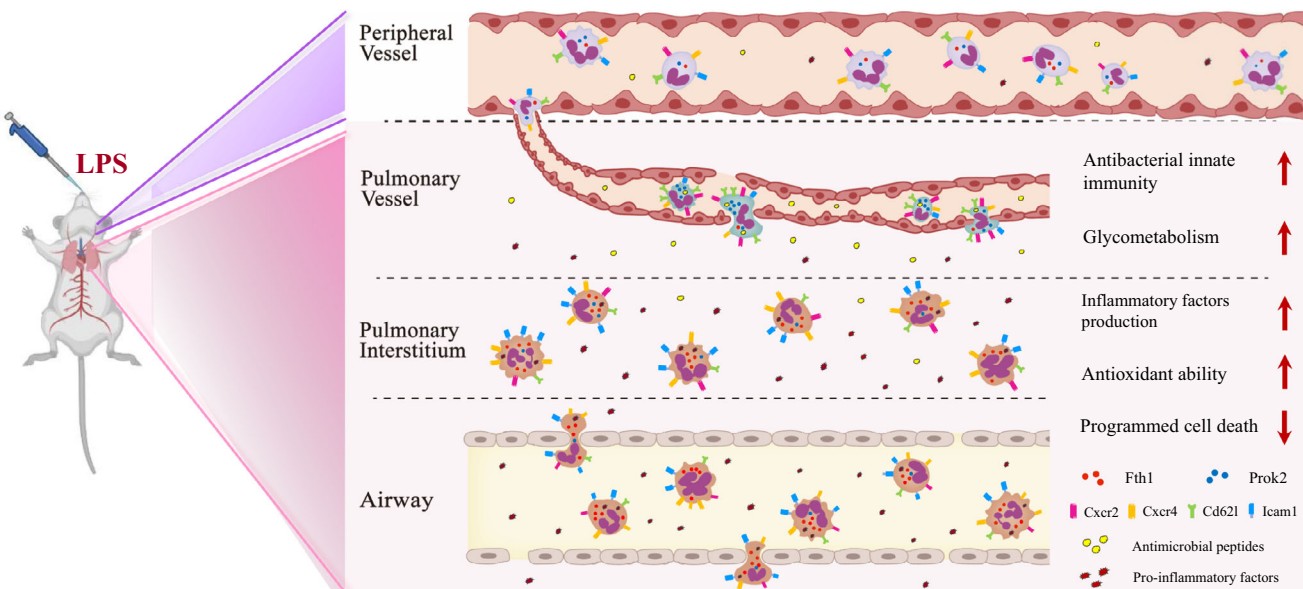

**Fig. 10 | Proposed mechanisms of two neutrophil subsets in lung tissue of ALI mice.** When neutrophils migrate from systemic circulation to the pulmonary vasculature in ALI mice, the effect of mechanical deformation and the characteristics of the microenvironment make them more adaptive to defense against pathogens (Prok2hi Neu). During subsequent trans-endothelial and -epithelial movement, large numbers of neutrophils develop into the Fth1hi subset, with increased resistance, delayed apoptosis and greater production of inflammatory mediators. Fth1hi Neu exhibited lower expression of L-selectin (CD62L) and C-X-C motif receptor 2 (Cxcr2), and higher expression of Cxcr4 and intercellular adhesion molecule-1 (Icam1). Some figure elements were created with BioRender.com.

variety of pathological conditions, which hinders the disease specificity of this indicator to a certain extent. Furthermore, underlying molecular mechanisms remain largely unknown regarding how elevated serum ferritin may lead to poor outcomes of certain diseases. We identified a potential clinical predictor from BALF samples rather than serum, which may better reflect the local pathological changes within the lungs. We supposed that increased Fth1 and decreased Prok2 expression in BALF neutrophils from ARDS patients exhibited defective ROS production, phagocytosis and apoptosis, while increased inflammation, which tended to poor prognosis (Supplementary Fig. 10). Thus, our study could offer improved tissue specificity for prognosis, though this possibility awaits confirmation with a more extensive and diverse sample size. We also determined that the poor outcomes of pulmonary infection may involve the prolonged retention of Fth1hi neutrophils within the lungs, which further supports and extends our understanding of ferritin in the development of pulmonary disease.

Collectively, we identified two transcriptional and functional heterogenous neutrophil subsets within LPS-induced ALI mice lung using the scRNA-seq technique. Notably, these subsets display distinct intra-pulmonary distributions and differences in locally orchestrated inflammation throughout the development course of ALI, which is mediated under the regulation of IL-10. We also determined that the abundance of BALF Fth1hi neutrophils provides a potential biomarker for predicting worse outcomes of pulmonary infection, though further studies are needed to support this possibility.

## Methods
### Study approval
All animal experiments were performed according to the guidelines of Shanghai Committee for Accreditation of Laboratory Animal, and the research protocol was approved by the Laboratory Animal Research Center Review Board of Tongji University (Permit Number: TJBB03721106) (Shanghai, China). All surgery was performed under pentobarbital sodium anesthesia with efforts to minimize animal suffering. A total of 24 participants from Shanghai East Hospital were enrolled for acquiring BALF samples. The study was approved by the ethics commission of Shanghai East Hospital, China (Permit Number: 2019YYS138), and informed consent was obtained from all participants, including publishing information that may identify individuals. All participants received the waivers of their medical expenses for compensations.

### LPS-induced acute lung injury (ALI) mouse model
B6.129P2-Il10tm1/Nju (*Il-10*[−/−]) and control (C57BL/6JNju; WT) mice (both male and female, 6-8 weeks old) were obtained from the Model Animal Research Center of Nanjing University (Jiangsu, China) and bred in Tongji University. Male C57BL/6 mice (6-8 weeks old) were purchased from Shanghai Laboratory Animal Co. Ltd., (Shanghai, China). The mice were housed under specific pathogen-free conditions and in a standarded light-dark cycle with food and water ad libitum. LPS (10 mg/kg; Sigma-Aldrich, USA) or equal volumes of PBS were intranasally administered to *Il-10*[−/−] or WT mice. Recombinant mouse IL-10 (rmIL-10; Biolegend, USA) was administered simultaneously with LPS challenge at a concentration of 45 µg/kg, with PBS used as negative control. Mice were sacrificed 6 h, 12 h, 24 h, 4 d, or 7 d post-exposure for subsequent sequencing and inflammation analysis. For survival analysis, mice were under a lethal dose (25 mg/kg) of LPS challenge to induce severe ALI. The mouse survival in each group was recorded every day for 7 days.

### Single-cell isolation of lung tissues
Single-cell suspensions from lung tissues were generated according to published protocols with minor modifications[49,50]. To minimize any potential effect on transcriptional profiles due to enzymatic digestion and cell sorting, we chose a protocol with a relatively short dissociation incubation period and maintained cell suspensions on ice for all subsequent steps. Mice were sacrificed by cervical dislocation after anesthetization with pentobarbital sodium. After exposing the thoracic cavity, the lungs were removed, minced and transferred into a 6-well plate containing enzyme solution (Collagenase Type II (450 U/mL), Elastase (4 U/mL) and DNase (1000 U/mL)). After 30 min incubation at 37 °C, DMEM-F12 medium was added to stop the enzymatic reaction. Subsequently, the cell suspensions were filtered through a 100 µm

strainer, transferred to a centrifuge tube, and spun at 500× *g* for 5 min. Red blood cell lysis buffer (Sigma-Aldrich, USA) was used to remove the erythrocytes. Living cells (>95%) were counted on a hemacytometer using the trypan blue exclusion method.

## ScRNA-seq library construction with the 10× Genomics platform

Gel Bead-In-Emulsions (GEMs) were generated by combining barcoded Single Cell 3′ Gel Beads, Master Mix, and Partitioning Oil (10× Genomics, USA) with the cells on a microfluidic chip. To achieve single cell resolution, the cells were delivered at a limiting dilution, such that the majority (~90 − 99%) of generated GEMs contained no cells, while the remainder largely contained a single cell. Immediately after generation of a GEM, the Single Cell 3′ Gel Bead was dissolved, and any co-partitioned cells were lysed. Upon dissolution, primers containing: (i) an Illumina R1 sequence (read 1 sequencing primer), (ii) a 16 nt 10× Barcode, (iii) a 10 nt Unique Molecular Identifier (UMI), and (iv) a poly-dT primer sequence were released and mixed with the cell lysate and Master Mix (Illumina, USA). Subsequent incubation of the GEMs produced barcoded, full-length cDNA from poly-adenylated mRNA.

After incubation, the GEMs were disrupted and the pooled fractions were recovered. Silane magnetic beads were used to remove remaining biochemical reagents and primers from the post GEM reaction mixture. Full-length, barcoded cDNA was then amplified by PCR to generate sufficient mass for library construction. R1 (read 1 primer sequence) was added during GEM incubation. P5, P7, and R2 (read 2 primer sequence) were added during library construction via End Repair, A-tailing, Adaptor Ligation, and PCR. The Single Cell 3′ 16 bp 10× Barcode and 12 bp UMI were encoded in Read 1, while Read 2 was used to sequence the cDNA fragment. Sample index sequences were incorporated as the i7 index read. Read 1 and Read 2 were standard Illumina sequencing primer sites used in paired-end sequencing.

## ScRNA-seq data processing and analysis

Sequencing data were aligned to the mouse genome mm10 with Cell Ranger version 6.0 (10× Genomics, USA). The data were processed using the Seurat R package version 4.0 and Loupe Browser version 5.0.0[51]. Cells with less than 500 detected genes, more than 25% percent mitochondria genes or UMI <500 were excluded. For cell clustering, we first constructed a K-nearest neighbor (KNN) graph based on the Euclidean distance in Principal Component Analysis (PCA) space, and then refined the edge weights between any two cells based on the shared overlap in their local neighborhoods (Jaccard distance). We further applied modularity optimization technique-SLM[52], to iteratively group cells together, with a goal of optimizing the cluster efficiency. t-SNE was used to visualize cells with similar local neighborhoods in high-dimensional space together in low-dimensional space. Differentially expressed genes were identified using a likelihood-ratio test for single-cell gene expression, as implemented by the Seurat's v6 FindAllMarkers function. Genes with *p* < 0.01 and fold change >1.5 were included in cluster marker signatures. Fisher's test was conducted to calculate whether the GO terms and KEGG pathway sets were significantly enriched in the target gene list, and P values were further calculated.

## SMART-Sequencing

Due to the relatively low abundance of airway and blood neutrophil subsets, we adopted the SMART-Seq v4 Ultra Low Input RNA Kit (Clontech, 635026, Japan) to prepare low-input libraries for the subsequent bioinformatics analysis. We used extracted poly-A RNA as template and oligo (dT) as the primer to synthesis the first strand using SMARTScribe Reverse Transcriptase (Illumina, USA). The sequences were further expanded using SMARTer Oligonucleotide as the template and amplified to generate the cDNA libraries. The primary libraries were then purified and size assessed using the Agilent 2100 Bioanalyzer (USA). A quality-control qualified library was amplified

using the PCR method to prepare a sequencing library. Sequencing was performed on the Illumina Novaseq™ 6000 (USA) platform using a 2 × 150 bp paired-end sequencing protocol.

Cutadapt software was used to remove the reads with adaptor contamination. After removal of the low quality or undetermined bases, HISAT2[53] software was used to map reads to the genome. The mapped reads of each sample were assembled using StringTie[54]. The expression levels of all transcripts were estimated as FPKM (total exon fragments/mapped reads (millions) x exon length (kB)). The differentially expressed mRNAs with fold change >2 or fold change <0.5 and *p* value < 0.05 were selected by edgeR, and the GO terms and KEGG Pathways of differentially expressed mRNAs were annotated.

## BALF and serum acquisition for local and systemic inflammation evaluation

After anesthetization, mice underwent disinfection and tracheotomy. Ice-cold PBS (0.8 mL) was slowly injected through the trachea, followed by carefully withdraw. The volume of the retrieved fluid (BALF) was >70% of the injected volume. BALF samples were centrifuged at 140 × g for 10 min at 4 °C to collect BAL inflammatory cells. Total cell counts were determined using a hemocytometer with the trypan blue dye exclusion test. Aliquots of BALF were fixed on glass slides via cytocentrifugation and further stained with Wright-Giemsa (Baso Diagnostics, China). Afterwards, differential cell counts were performed under a microscope at 400× magnification for at least 200 cells in a blinded fashion. Blood (0.5-0.8 mL) was collected by cardiac puncture via a syringe and transferred to a 1.5-mL Eppendorf Micro Test (EP) tube. After standing at room temperature for at least 1 h, the blood samples were centrifuged at 2380 × g for 10 min. BALF supernatants and serum were then aliquoted and stored at −80 °C for further experimentation. The levels of IL-6 (Invitrogen, 88-7064-88, USA), TNF (Invitrogen, 88-7324-88, USA), IL-12p70 (Invitrogen, 88-7121-88, USA), KC (MultiSciences Biotech, EK296/2-96, China), G-CSF (Multi-Sciences Biotech, EK269/2-96, China), IFN-γ (MultiSciences Biotech, EK280/3-96, China), IL-1β (MultiSciences Biotech, EK201B/3-96, China), IL-17A (MultiSciences Biotech, EK217/2-96, China) and IL-10 (Abcam, ab255729, UK) both in BALF and serum, were quantified using ELISA kits according to the manufacturer's instructions. The total BALF protein concentration was determined using a BCA-protein quantification assay kit (Beyotime, P0010, China).

## Histological and immunologic analyses of lung injury and neutrophil activation

For histological analyses, the left lung lobes were removed without collecting BALF and were fixed in 10% neutral formaldehyde and embedded into paraffin for tissue sectioning. The embedded waxes were cut into 5-µm slices and flattened on a glass slide, followed by hematoxylin-eosin (H&E) staining, dehydration and mounting. Evaluation of lung injury was performed by two independent investigators who were blinded to the samples using a reliable scoring system[55]. To assess the severity of lung damage, six randomly chosen fields were evaluated according to five independent variables: neutrophils in the alveolar space, neutrophils in the interstitial space, hyaline membranes, proteinaceous debris filling airspaces and alveolar septal thickening. Each variable was scored with 0, 1 or 2 according to the severity, with differential weighting according to the relevance of these variables. The sum of the weighted variables was averaged according to the number of determined fields, yielding a final score between 0 and 1. The slides were also immune-stained with anti-Ly6G (ab25377, 1:100), anti-MPO (ab20867, 1:1000), anti-NE (ab68672, 1:100) and anti-FHC (Fth1; ab65080, 1:200) antibodies (all from Abcam, UK) to determine the accumulation and activation status of neutrophils. Images were acquired on a Nikon Eclipse C1 microscope (Nikon, Japan) and processed with the ImageJ software (version 1.44p, National Institutes of Health, USA).

## Immunofluorescence assay of Fth1 and Prok2 expression in neutrophils

To identify Fth1$^{hi}$ Neu and Prok2$^{hi}$ Neu in mouse lung tissues, left lung lobes were collected without bronchoalveolar lavage and fixed with 10% neutral formaldehyde. Samples were embedded into paraffin and cut (4-μm-thick sections) for immunofluorescence staining. Lung sections were deparaffinized and rehydrated, followed by boiling for 20 min in a 10-mM sodium carbonate buffer for antigen retrieval. After permeabilization in 0.5% triton X-100/PBS solution, tissue slides were incubated with blocking buffer (Sigma-Aldrich, USA) at room temperature at 1 h and stained with rabbit anti-FHC (Fth1; 1:200), anti-Prok2 (Zen Bio, 506705, 1:500, China) and rat-anti-Ly6G (1:100) antibodies for 1 h. Afterward, the samples were washed with PBS and incubated with secondary goat anti-rabbit IgG antibody conjugated with Cy3 (Servicebio, GB21303, 1:300, China) and goat anti-rat IgG antibody conjugated with FITC (Servicebio, GB22302, 1:100, China) in blocking buffer containing DAPI (1:1000) for 2 h in the dark. Finally, the sections were mounted with 10-μl ProLong Antifade reagent (ThermoFisher, USA) and stored at room temperature in the dark overnight. Images were acquired using a fluorescence microscope (Nikon Eclipse C1, Japan) and Imaging system (Nikon DS-U3), and then were processed with ImageJ software.

## Isolation of neutrophils from BALF and peripheral blood

Neutrophils were isolated using Percoll gradient separation. Briefly, BALF cells were washed with pre-cold PBS (200× $g$, 4 °C, 5 min) and resuspended in 4-mL α-MEM medium overlaid by Percoll gradients (82%/65%/55%). The lower band was collected and washed after centrifuged at 590× $g$, 4 °C for 30 min. Erythrocyte lysis was performed with Pharm Lysis buffer (BD Bioscience, Canada), followed by centrifugation as recommended by the manufacturer.

For blood neutrophils, similar methods were employed. After anesthetization, cardiac puncture with a sodium citrate-coated syringe was performed to collect 0.5–0.8 mL blood per mouse, which was transferred to a 15-mL falcon tube with sodium citrate. Erythrocytes were immediately eliminated by the addition of Pharm Lysis buffer. The cell pellets were then resuspended in 2-mL α-MEM medium and carefully overlaid by the three-layer Percoll gradients. After gradient separation, a yield of $5 \times 10^5$ cells per mouse with viability ≥95% was obtained.

## Detection of neutrophil apoptosis by TUNEL and flow cytometry

The presence of apoptotic neutrophils in lung sections was examined using a TUNEL assay kit (Roche Diagnostics, 12156792910, 11684795910, USA). In brief, the left lungs were fixed in 10% neutral formaldehyde, embedded in paraffin and sectioned at 5 μm according to standard histology procedures. After dewaxing, rehydration and equilibration in Tris Buffered Saline, the samples were digested with proteinase K at room temperature for 20 min. Then, the sections were incubated with a mixture of TdT enzyme and fluorescently labeled nucleotides, followed by examination under a fluorescence microscopy. The apoptosis rate is represented by the percentage of TUNEL-positive cells.

Polychromatic flow cytometry was also performed to identify the apoptosis rate of neutrophils in the BALF. The cells were washed in PBS and stained with a cocktail of antibodies against phycoerythrin (PE)-Annexin V (BD Pharmingen, 559763, USA) and fluorescein isothiocyanate (FITC)-Ly6G (BD Pharmingen, 551460, 1:200, USA). After incubation for 15 min at room temperature in the dark, the cells were washed and stained with 7-Amino-Actinomycin (7-AAD) (BD Pharmingen, 559763, USA). The samples were then analyzed on a flow cytometer (BD Accuri C6, USA) within 1 h.

## Observation of neutrophil mitochondrial morphology by transmission electron microscopic (TEM)

For TEM imaging of neutrophil mitochondria, lung tissues were isolated and cut into 1-2 mm³ cubes. After fixation in 2.5% glutaraldehyde buffer for 24 h at 4 °C, the samples were washed with PBS, post-fixed in 1% osmium tetroxide for 1 h, dehydrated in graded ethyl alcohol solutions (30%, 50%, 70%, 90 and 100%), and embedded in epoxy resin. Ultrathin tissue sections (70 nm) were stained with lead citrate and uranyl acetate and observed on a transmission electron microscope (Hitachi HT7700, Hitachi, Tokyo, Japan). ImageJ software was adopted to manage the images and calculate mitochondrial morphological parameters according to the form factors (FF) of mitochondria = perimeter$^2$/(4π × area).

## Evaluation of mitochondrial respiratory chain (MRC) activity

MRC activity was detected with the MRC Complex I, III Activity Assay Kit (Solarbio Life Sciences, BC0510, BC3240 China) according to the manufacturer's instructions. The colorimetric assay evaluated activity at 340 nm (MRC complex I) and 550 nm (MRC complex III). The enzyme activity was calculated as nmol/min/mg protein.

## Ferric iron detection by Prussian Blue staining of lung tissues

Mouse lung samples were processed using a xylene-free method with isopropanol as the main substitute fixative. Iron deposition in mice lung tissues was visualized by Prussian blue staining, which appears as an intense blue. Lung sections were treated with an acid ferrocyanide solution to unmask ferric iron as Fe(OH)$_3$. The ferric iron and dilute potassium ferrocyanide react together to produce an insoluble blue compound, ferric ferrocyanide (Prussian blue). Briefly, tissues were cut into 3-μm slides, baked for 15 min at 60 °C, dewaxed by two xylene changes and rehydrated through transforming alcohol into water. After exposure to 2% aqueous potassium ferrocyanide and 2% HCl solutions for 30 min, the slides were washed with distilled water, counterstained with Van Gieson for 5 min, and blotted until dry. The samples were dehydrated in ascending grades of alcohol and cleared in xylene, followed by mounting with a cocktail containing distyrene, plasticizer and xylene.

## Human BALF sample acquisition

A total of 24 patients with pulmonary infection, lung cancer, pulmonary fibrosis and acute exacerbation of chronic obstructive pulmonary disease were enrolled, and 7 developed into ARDS according to Berlin criteria. Detailed demographic and clinical characteristics of the participants are provided in Supplementary Table 1. BALF was collected after informed consent within 48 h after admission. Briefly, 50 mL sterile isotonic saline was instilled into the subsegmental bronchus of the pathological entity, followed by recovery averaging 42 mL (range, 38-45 mL) with no difference among the groups. The BALF was immediately filtered and centrifuged, and BALF cells were washed, counted and fixed in 10% neutral formaldehyde.

## Cell culture, lentivirus transduction and functional assay of HL-60 cell-derived neutrophils

The human promyelocytic leukemia (HL-60) cell line was purchased from iCell Bioscience Inc, (Shanghai, China) and cultured in Iscoves modified Dulbecco medium (IMDM) medium (Gibco, USA) supplemented with 20% FBS (Gibco, USA) and 1% antibiotic-antimycotic solution at 37 °C, 5% CO$_2$. The cells were differentiated toward granulocyte-like cells with 1.5% Dimethylsulfoxide (DMSO; Sigma, USA) for 5 d. Recombinant lentiviral vector of Fth1/Prok2 shRNA were designed and constructed by Zorin (Shanghai, China). HL-60 cells were infected with lentivirus-mediated shRNA (*Fth1/Prok2*) or negative control shRNA (NEGi) respectively for 12 h. The cells were collected to determine the interference efficiency by reverse transcription-polymerase chain reaction and immunoblotting assay. The target sequences were GCTTTGAAGAACTTTGCCAAA for *Fth1* shRNA and GCTCTGGAGTAGAAACCAA for *Prok2* shRNA.

For ROS analysis, HL-60 cell-derived neutrophils in each group were incubated with 2′, 7′-dichlorofluorescein diacetate (DCFH-DA) fluorescent probe (10 μM; Sigma, USA) in the dark for 30 min. Then the

cells were rinsed thoroughly and collected for flow cytometer (Attune NxT, Invitrogen, USA). Intracellular ROS level was quantified by the fluorescence intensity of DCFH-DA. As for the phagocytosis assay, differentiated HL-60 cells were incubated with phenol red-free medium containing 1 mg/mL pHrodo Green E. coli BioParticles conjugate (Invitrogen, USA) for 1 h at 37 °C, 5% $CO_2$. Afterwards, samples were collected for flow cytometry and the fluorescence intensity was analyzed for phagocytosis.

The migration assay of HL-60 cell-derived neutrophils was carried out in Transwell chamber (Corning, USA) with 8 μm porous membrane. Cells were seeded to the upper chambers at a density of $2 \times 10^5$ cells in 200 μL/well, and allowed to migrate towards the lower chambers with or without N-formyl-methionyl-leucyl-phenylalanine (f-MLP, a neutrophil chemoattractant) stimulation for 12 h. Migrated cells in the lower chambers were collected and quantified using a hemocytometer. Apoptosis of the cells was estimated following the above method. Besides, HL-60 cell-derived neutrophils from each group were also collected for immunoblotting assay. Anti-NLRP3 (Cell Signaling Technology, 15101, 1:1000, USA), anti- Caspase-1 (Abcam, ab179515, 1:1000, UK), anti-HO-1 (Cell Signaling Technology, 5853, 1:1000, USA), anti-Bax (Zen Bio, 380709, 1:1000, China), anti-FHC (1:1000), anti-Prok2 (Abcam, ab76747, 1:1000, UK) and anti-β-actin (Cell Signaling Technology, 3700, 1:1000, USA) antibodies were used as primary antibodies. Anti-rabbit IgG (Cell Signaling Technology, 7074, 1:1000, USA) and anti-mouse IgG (Cell Signaling Technology, 7076, 1:1000, USA) were the secondary antibodies. Uncropped scans of all blots were supplied in Supplementary Fig. 16.

### Statistics
With the exception of the scRNA-seq and Smart-Seq, experimental data were analyzed using two-tailed unpaired Student t test, one-way or two-way ANOVA followed by Bonferroni's post hoc test by GraphPad Prism software (version 8.0.1, GraphPad Software, Inc., San Diego, CA). Normal distribution of the samples was tested by Shapiro-Wilk test. Statistical analysis for survival curve was performed using Log-rank test. Correlation between variables were assessed using the Pearson's correlation test. Data are expressed as mean ± standard error (±SEM), and differences were considered statistically significant when $p < 0.05$. All experiments, except for the scRNA-seq and SMART-sequencing, were performed independently at least three times.

### Reporting summary
Further information on research design is available in the Nature Portfolio Reporting Summary linked to this article.

## Data availability
The raw data of single-cell RNA-seq generated in this study have been deposited in Genome Sequence Archive with accession ID CRA008837. The SMART-seq data are available under the accession ID CRA008856. The remaining data are available within the Article, Supplementary Information or Source Data files. Source data are provided with this paper.

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

## Acknowledgements

This work was supported by National Natural Science Foundation of China (grant number: 82000086), Shanghai Sailing Program (grant number: 20YF1440300) for Wei Gao, National Natural Science Foundation of China (grant number: 81870064, 82070086) for Qiang Li, National Natural Science Foundation of China (grant number: 82100089) for Kun Wang. We thank BioRender for providing fundamental materials of our scheme figures (Figs. 1a, 3a and 10 created with BioRender.com).

## Author contributions

W.G. and Q.L. is responsible for the content of the manuscript, including the data and analysis. They conceived and designed the study. K.W., M.W. and X.L. coordinated to carry out experiments with technical guidance from W.G. and Q.L. S.G., J.H., X.W., Q.G., W.X., J.S. and Y.H. coordinated to collect human samples from respiratory and critical care department of Shanghai East Hospital under the guidance from Q.L. W.G., K.W. and X.L. analyzed and interpreted the data and wrote the manuscript. All authors read and approved the final manuscript.

## Competing interests

The authors declare no competing interests.
