## [Peer Review File · Nature Communications]

Locally Orchestrated Fth1hi Neutrophils Aggravate Inflammation of Acute Lung Injury in an IL-10-Dependent MannerREVIEWER COMMENTS

Reviewer #1 (Remarks to the Author):

1) General summary and opinion about the principal significance of the study, its questions and findings:

It is an interesting original work analyzing neutrophils transcriptional and functional heterogeneity of pulmonary neutrophils in an ARDS mouse model (LPS-exposed) and in ARDS patients. Since neutrophils play a key role in progression and resolution of ALI in ARDS, the authors studied the phenotypic heterogeneity of pulmonary neutrophils (Neu) PMN. Using a murine LPS model of ALI, they identified two transcriptional and functional PMN subsets (by single cell sequencing) localized in distinct lungs compartment: intranasal LPS induced Fth1hi pro-inflammatory PMN subset inside and around airways with elevated inflammatory factor production, strong antioxidant and decreased apoptosis and regulated by IL-10. In addition, they show that ARDS patients have high ration of Fth1hi /Prok2hi PMN suggesting that Fth1hi PMN is a marker of poor outcome.

2) Specific major concerns essential to be addressed:

-In the introduction, only one publication on Neu subsets in ALI/ARDS was cited (ref: 6) but it seems that other works were done.

-Since neutrophil extracellular traps (NETs) were recently associated with lung inflammation severity, ARDS (and in particular in ARDS following SARS-Cov-2 infection), did the authors looked at NETs and if yes, did they observed there difference in NETs formation between the two neutrophil (Neu) subsets described here? It will be probably important to look that.

-Do the two Neu subsets described, correspond to pro and anti-inflammatory Neu?

-Which cells produce IL-10 in that context?

-May IL-28 (Interferon lambda) have the same role than IL-10? Discuss that

-Is Neu expression of IL-10R different after LPS and in WT and Il-10ko mice?

- Why BALF Neu from patients with pulmonary infections expressed higher Prok2 and lower Fth1 levels in comparison to ARDS patients? Is there only differences between pulmonary infections and worse outcomes of pulmonary infection? Clarify this.

- Is it possible that the same Neu subsets exist in other lung diseases such as lung fibrosis, COPD, or ARDS associated with worse outcomes of pulmonary infection such as COVID-19? Discuss that.

- Is it possible to measure/detect ferritin in BALF from LPS-exposed mice and/or ARDS patients as measured in serum in other pathologies as discussed?

- The authors might present -ferric iron deposition (staining with Prussian blue) in neutrophils from BAL of LPS-mice or ARDS patients

Minor comments:

-Define Fth and Prok in abstract (not only in conclusion)

-Line 74: "are thought to be impacted": be is missing

-Explain Fth1-eCKO1 mice

Reviewer #2 (Remarks to the Author):

Wang and colleagues identify in this manuscript 2 new subpopulations of neutrophils in lungs during ALI/ARDS, namely neutrophils with high expression of Fth1 around airways, and those with high expression of Prok2 in vessels. The Fth1high neutrophils displayed a proinflammatory phenotype and the expression of Fth1 increased with progression of acute lung injury. Fth1high-to-Prok2high neutrophil ratio was increased in patients with ARDS and suggested to aggravate lung inflammation. The study provides interesting and comprehensive new insight into the emerging field of functionally distinct neutrophil subpopulations in health and disease, in this case lung inflammation. However, I

have a few suggestions to improve the manuscript.

A paragraph on Fth1 and Prok2 in the introduction would be helpful for the reader to put the functional relevance into context. What about potential functional differences of the human vs mouse proteins? Why is the IL10-KO + PBS control not shown for comparison?

Why were IL10 levels in BALF supernatant and serum not measured by ELISA? It would be important to see how IL10 behaves upon LPS treatment in the WT in their model and how that correlates over time with the other parameters analyzed.

Fig 1B-E, legend states „at different time points“, but only B shows a time curve; in C-E only one time point is shown, and the legend does not state which.

The neutrophil subtypes N1-N7 need to be introduced and discussed properly.

While the descriptive part of the subpopulations is very solid, the mechanistic insight of how IL10 regulates the distinct patterns/functions in these subpopulations is sparse. How exactly does IL10 control expression, apoptosis etc. How would neutrophils behave with these proteins (Fth1, Prok2) deleted? Experiments using depleted neutrophil cell lines such as HL60 could provide more mechanistic details! How about responsiveness to KC (highly overexpressed after 6h without IL10, Fig 1G) in the different subsets, which could affect the migration and localization pattern?

What is the relevance of the different transcriptional kinetics with and without IL10? This should at least be better discussed.

Are the newly described neutrophil subtypes somehow related to the ones described by the Kubes group that are retained in capillaries for emergency defense? This should be discussed. Also in this respect, do these new neutrophil subtypes show differences in phagocytic ability? It would be nice to see data on neutrophil effector functions related to the outcome of ALI/ARDS.

13 patients overall (pulmonary infection and cancer; and 8 for venous blood, 4 healthy + 4 ARDS) is not a lot to draw meaningful conclusions. The data appear preliminary for now. Is there a chance to increase the n in a reasonable time frame?

I could not read Suppl. Figures 7, 10 and 11; and Fig 3G, 4C, 8; and thus missed maybe some important information. They need to be presented in a more reader-friendly way.

Minor comments:

Results in Fig 3 need more detailed descriptions in the text.

Methods: I think it is enough to mention the supplementary methods once and not state it for all assays.

Please provide a reference for the “reliable scoring system previously published” (Suppl. Method line 129).

Statistics: Has normal distribution of data been tested? If so, what test has been used?

Discussion: I think it is too early to speak of a new nomenclature system. The findings need to be reproduced and generalized first (valid also in other organs during other diseases etc etc) to introduce new systematic names for neutrophil subpopulations. In general, the new findings should be discussed in more detail, i.e. putting the data in context with the existing literature on known neutrophil subsets in different pathophysiological conditions. In this respect, the discussion remains somewhat superficial.

Reviewer #3 (Remarks to the Author):

Manuscript by Kun Wang and colleagues describes two novel neutrophil subsets (Fthhi and Prok2) in a murine model of ARDS (LPS exposure) and their regulation by IL-10. Authors utilized a well-established murine model of ARDS and utilized IL-10 deficient mice as well as some human BAL and serum samples. Manuscript is well written, and figures are well presented. The study question is worth perusing and described experiments are quite straight forward and of limited novelty. However, significant concerns exist (noted below) about experimental design, data analysis, data interpretation and rigor of the findings.

Major concern:

One major concern is the lack of control group for PBS exposed IL-10 KO mice throughout the

manuscript. This is necessary control group which need to be presented for these studies. Statistical analyses are not well described and appear to be incorrectly chosen e.g., when looking for genotyping specific outcome with or without LPS exposure, then one way ANOVA is not suitable test for this purpose. Same comment stand for most of the data presented in this manuscript. It appears authors performed scRNA seq on just 1-2 mice. If this is the case these analyses don't have enough rigor.

Human data is from a very small patient population and is presumably not well powered.

Imaging data is presented as intensity per neutrophil which while helpful could bias the outcome when not considering the biological replicates/variability.

Survival data is not presented and need to be incorporated.

IL-10 inhibition as utilized (concurrent with LPS) is neither a prophylactic nor a therapeutic strategy. In order to validate its roles as either its dosing regimen need to be adjusted.

Minor Concern:

Methodology is presented for TEM evaluation of mitochondria, but images are not provided.

Response to the reviewer's comments:

Reviewer 1:

Major Comments

1. In the introduction, only one publication on Neu subsets in ALI/ARDS was cited (ref: 6) but it seems that other works were done.

Response: We thank the reviewer for the helpful comment and we are truly sorry for our previous negligence. By retrieving related researches, we have cited some primary results of recently published papers on Neu subsets in ALI/ARDS and added "...A subset of Ly6G⁺ neutrophils expressing high level of CD11b has been reported to retain within the pulmonary microvascular system under endotoxin infection and be capable of capturing disseminating pathogens¹. During the global pandemic of coronavirus disease 2019 (COVID-19), researchers identified the emergence of a low-density neutrophil population expressing intermediate levels of CD16 from the peripheral blood and bronchial lavage fluid (BALF) of COVID-19-associated ARDS patients^{2, 3}. Such type of neutrophils demonstrated enhanced phagocytosis, neutrophil extracellular trap (NET) formation, platelet activation, cytokine production, T-cell activation, and impaired degranulation^{4, 5}, thus were considered as a predictor of risk for worse outcomes for COVID-19 patients." in the "Introduction" according to the suggestion (Page 3-4).

2. Since neutrophil extracellular traps (NETs) were recently associated with lung inflammation severity, ARDS (and in particular in ARDS following SARS-Cov-2

infection), did the authors look at NETs and if yes, did they observe their difference in NETs formation between the two neutrophil (Neu) subsets described here? It will be probably important to look that.

Response: We thank the reviewer for bringing up this interesting question. To address this, recombinant lentiviral vectors of Fth1/Prok2 shRNA were designed and constructed by Zorin (Shanghai, China). Human promyelocytic leukemia (HL-60) cells were infected with lentivirus-mediated shRNA (Fth1/Prok2) or negative control shRNA (NEGi) respectively for 12 h. The cells were collected to determine the interference efficiency by reverse transcription-polymerase chain reaction (RT-PCR) and western blotting (WB) assay (**Figure R1A-B**). The target sequences were GCTTTGAAGAACTTTGCCAAA for Fth1 shRNA3 and GCTCTGGAGTAGAAACCAA for Prok2 shRNA2. Then the cells were differentiated toward granulocyte-like cells with 1.5% Dimethylsulfoxide (DMSO) and stimulated by phorbol 12-myristate 13-acetate (PMA), a NETs inducer. NETs release was analyzed using fluorescent microscopy, which showed that Fth1 depletion significantly increased while Prok2 knockdown reduced NETs formation compared with the control shRNA (**Figure R1C**). The results indicated that Fth1^{hi} neutrophils exhibited decreased NETs formation, which tended to be elevated in Prok2^{hi} neutrophils.

Figure R1. Fth1 and Prok2 modulate NETs formation in HL-60 cell-derived neutrophils. (A-B) Transfection of HL-60 cells with Fth1/Prok2 shRNA downregulated the mRNA (A) and protein (B) expression at 12 h compared with control shRNA (NEGi) treatment. Data are presented as mean \pm SEM. Statistical analysis was performed using one-way ANOVA followed by Bonferroni's post hoc test. N = 3 per group from 3 independent experiments. * $p < 0.05$, ** $p < 0.01$, **** $p < 0.0001$, ns: not significant. (C) Representative fluorescence images of Propidium Iodide (PI)-stained HL-60 cell-derived neutrophils with or without PMA stimulation. White arrows

indicate NET-forming neutrophils. Scale bar: 40 μ m. **(D)** Frequency of NET-forming neutrophils assessed by fluorescent microscopy. Data are presented as mean \pm SEM. Statistical analysis was performed using two-tailed unpaired Student t test. N = 4 per group from 4 independent experiments. ***p < 0.001, ns: not significant versus unstimulated NEGi; #p < 0.01, ####p < 0.0001 versus PMA-stimulated NEGi.

3. Do the two Neu subsets described, correspond to pro and anti-inflammatory Neu?

Response: We thank the reviewer for bring up this significant question. Fth1^{hi} neutrophils identified in the present study has been proved to be essential for unrestrained pulmonary inflammation and an increased ratio of Fth1 to Prok2 expression may be a potential marker for the more severe form of pulmonary infection. Using the recombinant lentiviral vector of Fth1 shRNA and HL-60 cells mentioned above, we found that Fth1 deletion dramatically impaired chemotaxis with or without N-formyl-methionyl-leucyl-phenylalanine (f-MLP, a neutrophil chemoattractant) stimulation (**Figure R2A**) and inhibited the activation of inflammatory NLRP3/Caspase-1 signaling (**Figure R2B**) compared with the control shRNA. Thus, we consider Fth1^{hi} Neu have some pro-inflammatory properties. As for Prok2^{hi} neutrophils, elevated genes associated with innate immunity often relates with overwhelming inflammation. With the shRNA lentiviral vector of Prok2, we discovered that Prok2 deficient neutrophils also exhibited functional defects in chemotaxis (**Figure R2A**), but had no significant impact on NLRP3/Caspase-1

pathway (**Figure R2B**). Therefore, we think the evidence is not enough to classify these two subsets into pro- and anti-inflammatory categories.

Figure R2. Effects of Fth1 and Prok2 on inflammation in HL-60 cell-derived neutrophils. (A) Fth1/Prok2 depletion inhibited chemotaxis of HL-60 cell-derived neutrophils compared with NEGi. Data are presented as mean ± SEM. Statistical analysis was performed using two-tailed unpaired Student t test. N = 4 per group from 4 independent experiments. *p < 0.05, ns: not significant versus unstimulated NEGi; #p < 0.05 versus f-MLP-stimulated NEGi. (B) Immunoblotting verifying the effects of Fth1/Prok2 on NLRP3 and cleaved Caspase-1 level for inflammasome activation.

4. Which cells produce IL-10 in that context?

Response: We thank the reviewer for bringing up this helpful question. IL-10 is one of the most important immunoregulatory cytokines with anti-inflammatory properties.

Although Th2 cells were the first identified cellular source of IL-10⁶, we now know that cells of both the myeloid and lymphoid lineages secrete IL-10 in response to different stimuli. This includes macrophages, monocytes, DCs, neutrophils, mast cells, eosinophils, and natural killer cells, in addition to CD4 and CD8 T cells and B cells⁷. Notably, some nonhematopoietic cells, including epithelial cells, are also able to produce IL-10⁸. In this study, we discovered that IL-10 mainly expressed in inflammatory cells, including neutrophils, monocytes/macrophages and T cells (**Figure R3**).

Figure R3. Immunohistochemical images of murine lung sections stained with IL-10.

Scale bar in the upper panel: 50 μm ; scale bar in the below panel: 20 μm .

5. May IL-28 (Interferon lambda) have the same role than IL-10? Discuss that

Response: We thank the reviewer for bring up the question. IL-28, commonly classified as type III interferons (IFN- λ), is also classified as IL-10 family cytokines based on their structural similarity, common receptor usage and down-stream signaling⁹. IL-28 signaling was traditionally recognized as a first-line defense against virus infection and concentrated at anatomic barriers like airway epithelial cells¹⁰. Recently, Blazek et al. reported that IL-28 could also exerts the anti-inflammatory activity by restricting recruitment of IL-1 β -expressing neutrophils, thus reversing the development of collagen-induced arthritis¹¹. However, Luo et al. demonstrated that administration of recombinant IL-28 aggravated mortality, facilitated bacterial dissimulation and limited neutrophil recruitment, in the model of sepsis induced by cecal ligation and puncture¹². Therefore, we estimated whether IL-28 have the similar role as IL-10 in the pulmonary inflammatory scenario using our mouse model. Recombinant murine IL-28A/IL-28B protein (2 μg , R&D systems) was administered simultaneously with LPS stimulation, followed by lung inflammation and tissue damage evaluation 24 h later. PBS was delivered in a similar fashion as control vehicle. As shown in **Figure R4A-B**, IL-28 did not affect the total cell number, especially neutrophils, as well as IL-6, G-CSF, and total protein concentration in the BALF of ALI mice. In parallel, the lung injury in ALI mice was assessed by histological features from the pathological tissue slides of murine lungs.

It was clearly observed that IL-28 administration could not alleviate LPS-induced tissue injury (**Figure R4C**). Furthermore, mice lung and BALF cell sections were stained with Fth1 and Ly6G, which indicated that Fth1 expression in ALI neutrophils was unaffected by IL-28 treatment (**Figure R5A-B**). Collectively, these data demonstrated that IL-28 differed from IL-10 in the pulmonary inflammatory environment in our study.

Figure R4. Recombinant IL-28A has no effect on pulmonary inflammation in LPS-induced ALI murine model. (A-B) IL-28 did not affect the total cell number, especially neutrophils and macrophages (A), as well as IL-6, G-CSF, and total protein concentration (B) in the BALF of ALI mice. (C) Histological images of the hematoxylin-eosin (H&E)-stained lung sections from distinct groups. Scale bar in the upper panel: 100 μm ; scale bar in the below panel: 40 μm . Data are presented as mean \pm SEM. Statistical analysis was performed using one-way ANOVA followed by Bonferroni's post hoc test. N = 5 per group from 3 independent experiments. ns: not significant.

Figure R5. Recombinant IL-28A has no effect on Fth1 expression in ALI neutrophils. (A-B) Representative images of anti-Fth1/anti-Ly6G immunofluorescence-stained lung tissues (A) and airway inflammatory cells (B) of ALI mice with or without recombinant IL-28A administration. Nuclei were stained with DAPI (4',6-Diamidino-2-phenylindole), displayed in blue. Scale bar: 40 μm.

6. Is Neu expression of IL-10R different after LPS and in WT and IL-10ko mice?

Response: We thank the reviewer for bring up the question. By using of immunofluorescence assay, we determined that IL-10R (Abcam) expression in neutrophils exhibited no difference between WT and IL-10^{-/-} mice. However, it was dramatically elevated after LPS challenge in WT mice, but decreased in IL-10^{-/-}-LPS group (**Figure R6**).

Figure R6. IL-10R expression in neutrophils of WT and IL-10^{-/-} mice with or without LPS challenge. (A) Representative images of anti-IL-10R/anti-Ly6G immunofluorescence-stained lung tissues from distinct groups. (B) The mean fluorescence intensity was calculated from more than 60 cells in each group from 5

mice. Nuclei were stained with DAPI. Data are presented as mean \pm SEM. Statistical analysis was performed using one-way ANOVA followed by Bonferroni's post hoc test.

** $p < 0.01$, **** $p < 0.0001$, ns: not significant.

7. Why BALF Neu from patients with pulmonary infections expressed higher Prok2 and lower Fth1 levels in comparison to ARDS patients? Is there only differences between pulmonary infections and worse outcomes of pulmonary infection? Clarify this.

Response: We are truly sorry for not making it clear in the previous manuscript and appreciate the reviewer for pointing it out. In the present study, we observed increased Fth1 and decreased Prok2 level in BALF neutrophils from ARDS patients compared to patients without ARDS. We deduced that attributed to the elevated proportion of Fth1^{hi} neutrophils identified in our study with a predominate 'senescence phenotype', and played an essential role in the poor prognosis of ALI modulated by IL-10. To further verify this, we employed the recombinant lentiviral vector of Fth1/Prok2 shRNA used in Figure R1. As shown in **Figure R7**, Fth1-depleted neutrophils exhibited function defects in anti-oxidation (**Figure R7A**), anti-apoptosis (**Figure R7C**) and chemotaxis (**Figure R7D**), while Prok2-deficient neutrophils defected in ROS production (**Figure R7A**), phagocytosis (**Figure R7B**) and chemotaxis (**Figure R7D**). The data indicated that increased Fth1 and decreased Prok2 expression in BALF neutrophils from ARDS patients exhibited defective ROS production, phagocytosis and apoptosis, while increased chemotaxis, which tended to poor prognosis. Besides, the difference of

Fth1/Prok2 expression may not only be between pulmonary infections and worse outcomes of pulmonary infection, but also between the patients with malignant tumor, which is also a risk factor of ARDS, and ARDS since the BALF neutrophils from a couple of cancer patients had also been investigated (**Figure 9**).

Figure R7. Effects of Fth1/Prok2 depletion on ROS generation, phagocytosis, apoptosis and chemotaxis in HL-60 cell-derived neutrophils. (A-D) Fth1-depleted neutrophils exhibited function defects in anti-oxidation (A), anti-apoptosis (C) and chemotaxis (D), while Prok2-deficient neutrophils defected in ROS production (A), phagocytosis (B) and chemotaxis (D) of HL-60 cell-derived neutrophils compared with NEGi. Data are presented as mean \pm SEM. Statistical analysis was performed using two-tailed unpaired t Student test. N = 4 per group from 4 independent experiments. *p < 0.05, ns: not significant (A-C). *p < 0.05, ns: not significant versus unstimulated NEGi; #p < 0.05 versus f-MLP-stimulated NEGi (D).

8. Is it possible that the same Neu subsets exist in other lung diseases such as lung fibrosis, COPD, or ARDS associated with worse outcomes of pulmonary infection such as COVID-19? Discuss that.

Response: We thank the reviewer for bring up the question and we believe it is worth in-depth study. A growing body of literatures demonstrated that cellular ferroptosis and Fth1 were involved in the pathogenesis of various pulmonary diseases including lung fibrosis, COPD and ARDS, with distinct effects. Yuan et al. reported that the expression of Fth1 was significantly down-regulated in the silica-induced pulmonary fibrosis¹³ while Wang and colleagues found that Fth1 was up-regulated in COPD patients¹⁴. Recently, Jiao et al. discovered that Fth1 level was decreased in mice lung after LPS stimulation¹⁵. However, the cellular targets of these researches were mainly epithelial

cells or macrophage; to our knowledge, this is the first study that investigated the relationship between Fth1 and neutrophil and its clinical implication in ALI/ARDS.

As for Prok2^{hi} subset, Sasaki and colleagues revealed that these neutrophils augmented the proliferation of breast cancer cells in vitro and enhanced the lung metastasis formation in vivo¹⁶. Recently, Yu et al. reported that serum Prok2 concentration was dramatically decreased in the patient with sepsis and septic shock compared with healthy controls and it could enhance the phagocytic and bactericidal functions of macrophage, demonstrating a central role in alleviating sepsis-induced death by regulating the function of macrophage¹⁷. However, whether this effect was mediated by neutrophils remained unclear.

Recently, we collected the BALF of another 11 inpatients from our department, among which 2 were pulmonary fibrosis, 2 were acute exacerbation of chronic obstructive pulmonary disease (AECOPD), and 5 developed into ARDS within 48h after admission. For the BALF neutrophils, we also observed increased Fth1 and decreased Prok2 expression in the ARDS patient (**Figure R8C**) compared to pulmonary fibrosis (**Figure R8A**), as well as AECOPD (**Figure R8B**). However, the samples were quite small so it needs to be continuously investigated in the future research.

Figure R8. Fth1 and Prok2 expression in BALF neutrophils from different patients. (A-C) Representative fluorescence images of anti-Fth1/anti-Ly6G and anti-Prok2/anti-Ly6G-stained airway neutrophils from patients with pulmonary fibrosis (A), AECOPD (B) and ARDS (C). The nuclei were stained with DAPI, displayed in blue.

9. Is it possible to measure/detect ferritin in BALF from LPS-exposed mice and/or ARDS patients as measured in serum in other pathologies as discussed?

Response: We thank the reviewer for bring up the question. Using ELISA assay, we discovered that BALF ferritin (Abcam) level was similar in WT and IL-10^{-/-} mice. However, it dramatically elevated after LPS challenge in WT mice, and increased further in IL-10^{-/-}-LPS group (**Figure R9A**). Moreover, human BALF ferritin (Abcam) was also detected and found to be elevated in ARDS patients compared with control participants (**Figure R9B**). The result needs to be further confirmed in larger cohort.

Figure R9. (A-B) BALF ferritin level in distinct mice groups (A) and patients with or without ARDS (B). Data are presented as mean \pm SEM. Statistical analysis was performed using one-way ANOVA followed by Bonferroni's post hoc test (A) and two-tailed unpaired Student t test (B). N \geq 5 per group from 3 independent experiments. *p < 0.05, **p < 0.01, ns: not significant.

10. The authors might present -ferric iron deposition (staining with Prussian blue) in neutrophils from BAL of LPS-mice or ARDS patients

Response: We thank the reviewer for bring up the question. By using Prussian blue staining of BALF cells from ALI mice and ARDS patients, we found striking prominent

ferric iron deposition (shown in brown) in neutrophils of LPS-exposed IL-10^{-/-} mice (**Figure R10A**) and ARDS patients (**Figure R10B**), compared to WT mice and patients without ARDS, respectively. The results are consistent with the data in the manuscript.

Figure R10. (A-B) Microscopic ferric iron deposition (shown in brown) in BALF neutrophils from distinct mice groups (A) and patients with or without ARDS (B).

Minor Comments

1. Define Fth and Prok in abstract (not only in conclusion)

Response: We appreciate the comment from the reviewer, which are very helpful to improve our manuscript. As suggested, we have added the definition of Fth1 and Prok2 and revised the abstract as “... neutrophils highly expressed *Prok2* (Prokineticin 2), defined as Prok2^{hi} Neu, located within pulmonary vessels, and neutrophils highly expressed *Fth1* (Ferritin Heavy Chain 1), referred to as Fth1^{hi} Neu, distributed widely in or around airway.” accordingly (Page 2).

2. Line 74: “are thought to be impacted”: be is missing

Response: We apologize for the clerical mistake in the previous manuscript and really thank the reviewer for pointing it out. We have revised the article accordingly (Page 4).

3. Explain Fth1-eCKO1 mice

Response: We thank the reviewer for bring up the question. The full name of Fth1-eCKO1 mice is C57BL/6-Fth1^{em1(flox)Smoc}, exon 2 of the Fth1 transcript was selected as the flox region, and em indicated endonuclease-mediated mutation, according to a previously published manuscript¹⁸ with the assistance of Shanghai Model OrganismsTM. CRISPR/Cas9 technique was adopted in the present project to establish *Fth1* conditional knockout flox mice. We injected guide-RNA, Cas9 mRNA, and vectors containing donor DNA into zygotes by micro-injection, and positively depleted mice were screen by polymerase chain reaction (PCR) and sequencing and bred. We now have used Fth1-eCKO1 mice to cross with transgenic S100A8-Cre mice, and obtained Fth1^{hi} Neu conditional knockout mice, thus to further validate our hypothesis.

Reviewer 2:

Major Comments

1. A paragraph on Fth1 and Prok2 in the introduction would be helpful for the reader to put the functional relevance into context. What about potential functional differences of the human vs mouse proteins?

Response: We appreciate the reviewer's suggestion. We have revised the introduction and added some brief description of Fth1 and Prok2 as “...One subset expressed high level of *Fth1* (Ferritin Heavy Chain 1), which is a cytosolic iron storage protein that plays a key regulator of ferroptosis, autophagy, oxidative stress and inflammation^{19, 20}. The other subset highly expressed *Prok2* (Prokineticin 2), a newly identified chemokines with a critical role in the immune system and likely participation in the pathogenesis of various inflammatory-related diseases^{21, 22}.” (Page 5).

As for the functional diversity between human and mice, we have demonstrated that Fth1 might have a direct role in anti-oxidant capacity and apoptosis resistance of Fth1^{hi} Neu in ALI mouse lungs. By use of recombinant lentiviral vector of Fth1 shRNA (Zorin, Shanghai, China) and human promyelocytic leukemia (HL-60) cell line, we further explored the function of Fth1 in human granulocyte-like cells. The target sequence for Fth1 shRNA3, which had the highest interference efficiency (**Figure R11A-B**), was GCTTTGAAGAACTTTGCCAAA. As shown in **Figure R11C-E**, Fth1-depleted neutrophils exhibited functional defects in anti-oxidation (**Figure R11C, E**) and anti-apoptosis (**Figure R11D, E**), which was consistent with the data in the mice.

Even though, we are totally agreed with the reviewer that this issue needs to be further elucidated.

Figure R11. Fth1 modulates ROS generation and cellular apoptosis in HL-60 cell-derived neutrophils. (A-B) Transfection of HL-60 cells with Fth1 shRNA downregulated the mRNA (A) and protein (B) expression at 12 h compared with control shRNA (NEGi) treatment. Data are presented as mean \pm SEM. Statistical analysis was performed using one-way ANOVA followed by Bonferroni's post hoc test. N = 3 per group from 3 independent experiments. ****p < 0.0001. (C-D) Fth1-depleted neutrophils exhibited function defects in anti-oxidation (C) and anti-apoptosis (D) of HL-60 cell-derived neutrophils compared with NEGi. Data are presented as mean \pm SEM. Statistical analysis was performed using two-tailed unpaired Student t test. N = 4 per group from 4 independent experiments. *p < 0.05. (E) Fth1 deficiency inhibited the expression of anti-oxidant protein, HO-1; while elevated the pro-apoptotic Bax protein level.

2. Why is the IL10-KO + PBS control not shown for comparison?

Response: We are truly sorry for not making it clear in the previous manuscript and appreciate the reviewer for pointing it out. We have revised **Figure 1** and **Figure 2** accordingly, which exhibited no significant difference of airway inflammation and lung injury between WT-PBS and IL-10^{-/-}-PBS control.

3. Why were IL10 levels in BALF supernatant and serum not measured by ELISA? It would be important to see how IL10 behaves upon LPS treatment in the WT in their model and how that correlates over time with the other parameters analyzed.

Response: We thank the reviewer for bring up the question. By using the ELISA assay, we discovered that IL-10 content was low in mice BALF supernatant, and it further reduced 24 h after LPS administration (**Figure R12A**). As for serum IL-10, it sharply elevated at 6 h after LPS challenge, followed by a dramatic decline in 12 and 24 h (**Figure R12B**). Interestingly, it was accompanied by the changes of pro-inflammatory cytokines IL-6, KC and TNF- α (**Figure 1F-H**), which reached a peak at 6 h in WT-LPS mice and further increased in IL-10^{-/-}-LPS mice. The data indicated that serum IL-10 elevation within 6 h might play a protective role by inhibiting early expression of pro-inflammatory cytokines in the ALI model. We appreciate the reviewer for pointing it out and have added the results as **Supplemental Figure 1** in our manuscript.

Figure R12. (A-B) IL-10 level in BALF supernatant (A) and serum (B) of ALI mice at different time points upon LPS treatment. Data are presented as mean \pm SEM. Statistical analysis was performed using one-way ANOVA followed by Bonferroni's post hoc test. N = 6 per group from 3 independent experiments. **p < 0.01.

4. Fig 1B-E, legend states “at different time points”, but only B shows a time curve; in C-E only one time point is shown, and the legend does not state which.

Response: We are truly sorry for not making it clear in the previous manuscript. We have revised **Figure 1C-E** and related legends according to the suggestion.

5. The neutrophil subtypes N1-N7 need to be introduced and discussed properly.

Response: We thank the reviewer for bring up the helpful question. We have added the highly expressed genes of N1-N7 in Results as “...including N1 (35.5%, highly expressed *Fth1* and *S100a8*), N2 (34.4%, highly expressed *Fth1* and *S100a8*), N5 (17.9%, highly expressed *Fth1* and *S100a8*), N6 (12.2%, highly expressed *S100a8* and *S100a9*) from IL-10^{-/-} ALI mice; and N3 (42.3%, highly expressed *Fth1* and *S100a8*), N4 (37.8%, highly expressed *Fth1* and *S100a8*), N7 (19.9%, highly expressed *S100a8*

and *S100a9*) from WT ALI mice...” (Page 9). Combine the supplemental Figure 8A-B, we consider similar gene expression pattern within N1-5 and N6-7, respectively.

6. While the descriptive part of the subpopulations is very solid, the mechanistic insight of how IL10 regulates the distinct patterns/functions in these subpopulations is sparse. How exactly does IL10 control expression, apoptosis etc. How would neutrophils behave with these proteins (Fth1, Prok2) deleted? Experiments using depleted neutrophil cell lines such as HL60 could provide more mechanistic details! How about responsiveness to KC (highly overexpressed after 6h without IL10, Fig 1G) in the different subsets, which could affect the migration and localization pattern?

Response: We appreciate the comment from the reviewer, which are very helpful to improve our manuscript. To address this, recombinant lentiviral vector of Fth1/Prok2 shRNA were designed and constructed by Zorin (Shanghai, China). Human promyelocytic leukemia (HL-60) cells were infected with lentivirus-mediated shRNA (Fth1/Prok2) or negative control shRNA (NEGi) respectively for 12 h. The cells were collected to determine their interference efficiency by reverse transcription-polymerase chain reaction (RT-PCR) and western blotting (WB) assay (**Figure R13A-B**). The target sequences were GCTTTGAAGAACTTTGCCAAA for Fth1 shRNA3 and GCTCTGGAGTAGAAACCAA for Prok2 shRNA2. Then the cells were differentiated toward granulocyte-like cells with 1.5% Dimethylsulfoxide (DMSO) for 5 days. As shown in **Figure R13**, Fth1-depleted neutrophils exhibited functional defects in anti-oxidation (**Figure R13C**), anti-apoptosis (**Figure R13E**) and chemotaxis (**Figure**

R13F), while Prok2 deficient neutrophils defected in ROS production (**Figure R13C**), phagocytosis (**Figure R13D**) and chemotaxis (**Figure R13F**). Besides, Fth1 deletion significantly inhibited the expression of anti-oxidant HO-1 and the activation of pro-inflammatory NLRP3/Caspase-1 signaling, while elevated pro-apoptotic Bax level compared with the control shRNA, which were not affected by Prok2 (**Figure R13G**). The data indicated that IL-10-elevated Fth1^{hi} neutrophils exhibited increased chemotaxis and inflammation while defective ROS production and apoptosis, which could explain the unrestrained pulmonary inflammation induced by LPS. Nevertheless, we think the conclusion needs to be continuously investigated in the future research. We thank the reviewer for pointing it out and have added the results as **Supplemental Figure 10** in our manuscript.

Figure R13. Effects of Fth1/Prok2 depletion on the immune function of HL-60 cell-derived neutrophils. (A-B) Transfection of HL-60 cells with Fth1/Prok2 shRNA downregulated the mRNA (A) and protein (B) expression at 12 h compared with control shRNA treatment. Data are presented as mean \pm SEM. Statistical analysis was performed using one-way ANOVA followed by Bonferroni's post hoc test. N = 3 per group from 3 independent experiments. * $p < 0.05$, ** $p < 0.01$, **** $p < 0.0001$, ns: not

significant. (C-F) Fth1-depleted neutrophils exhibited function defects in anti-oxidation (C), anti-apoptosis (E) and chemotaxis (F), while Prok2-deficient neutrophils defected in ROS production (C), phagocytosis (D) and chemotaxis (F) of HL-60 cell-derived neutrophils compared with NEGi. Data are presented as mean \pm SEM. Statistical analysis was performed using two-tailed unpaired Student t test. N = 4 per group from 4 independent experiments. *p < 0.05, ns: not significant (C-E). *p < 0.05, ns: not significant versus unstimulated NEGi; #p < 0.05 versus f-MLP-stimulated NEGi (F). (G) Immunoblotting verifying the effects of Fth1/Prok2 on anti-oxidant HO-1 and pro-apoptotic Bax expression, as well as NLRP3 and cleaved Caspase-1 level for inflammasome activation.

7. What is the relevance of the different transcriptional kinetics with and without IL10?

This should at least be better discussed.

Response: We thank the reviewer for bringing up this instructive question. The transcriptional regulation of IL-10 has been broadly explored, and a large body of literature documents reported that transcriptional factors, including c-Maf, Bhlhe40, Blimp-1, SMAD4 and AhR were involved in the regulation of IL-10 production^{23, 24, 25}. However, researches on global transcriptional kinetics alterations with and without IL-10 are quite scarce. Recently, Shirakawa and colleagues reported that IL-10 knockout significantly inhibited the tyrosine phosphorylation of STAT3, and suppressed the transcriptional activation of Spp1, thus further impaired the reparative macrophage polarization after myocardial infarction²⁶.

In the present study, we adopted single-cell differential expression analysis of transcriptional factors (TF) between WT and IL-10^{-/-} ALI mice lungs, and identified characteristic transcriptional kinetics with and without IL-10 (**Figure R14A**). The activity of Irf7, Jdp2, Ets2, Tgif1, Irf1, Cebpb, STAT1, and STAT3 were significantly higher, while Egr2, Ets1, Maff and Foxp1 were lower in lung cells of IL-10^{-/-} mice compared to WT mice after LPS challenge (**Figure R14B**). As for neutrophils, STAT3, STAT1, Cebpb, Ets2, Jdp2, Irf7 and Irf1 were significantly elevated in IL-10^{-/-} ALI mice (**Figure R14C**). These findings inspired us to further elucidate the impact on transcriptional kinetics of IL-10.

Figure R14. (A) Transcriptional kinetics of ALI mice lungs with and without IL-10.

(B-C) Transcriptional factors with different activities in lung cells (B) and neutrophils

(C) between IL-10^{-/-} and KO mice upon LPS challenge.

8. Are the newly described neutrophil subtypes somehow related to the ones described by the Kubes group that are retained in capillaries for emergency defense? This should be discussed. Also in this respect, do these new neutrophil subtypes show differences in phagocytic ability? It would be nice to see data on neutrophil effector functions related to the outcome of ALI/ARDS.

Response: We appreciate the reviewer's valuable comments. According to the suggestion, we have added the comments in Discussion as “...A previous study demonstrated that vascular neutrophils were rapidly arrested in the pulmonary microvasculature after bacterial stimulation²⁷. Kubes and colleagues further reported that pulmonary neutrophils sequestered within the capillaries after intravenous endotoxin challenge in a CD11b-dependent process and eliminated bloodstream pathogens¹. Although the transcriptome comparison is absent, Prok2^{hi} Neu identified in the present study seems to be closely related to capillaries-sequestered neutrophils defined by Kubes group on account of similar location and functional characteristics.” (Page 20-21). Moreover, with the recombinant lentiviral vector of Fth1/Prok2 shRNA, we demonstrated that Prok2 depleted neutrophils exhibited functional defects in ROS production (**Figure R13C**), phagocytosis (**Figure R13D**) and chemotaxis (**Figure R13F**), while Fth1 deficient neutrophils defected in anti-oxidation (**Figure R13C**), anti-apoptosis (**Figure R13E**) and chemotaxis (**Figure R13F**). Hence, it is reasonable to speculate that an increased ratio of Fth1 to Prok2 expression in BALF neutrophils from ARDS patients exhibited defective ROS production, phagocytosis and apoptosis, but increased chemotaxis, which tends to poor prognosis of ALI/ARDS.

9. 13 patients overall (pulmonary infection and cancer; and 8 for venous blood, 4 healthy + 4 ARDS) is not a lot to draw meaningful conclusions. The data appear preliminary for now. Is there a chance to increase the n in a reasonable time frame?

Response: We thank the reviewer for bring up the question and we believe it is worth in-depth study. Recently, we collected the BALF of 11 inpatients from our department, among which 2 were pulmonary fibrosis, 2 were acute exacerbation of chronic obstructive pulmonary disease (AECOPD), 2 were pulmonary infection, and another 5 developed into ARDS within 48h after admission. For the BALF neutrophils, we also observed increased Fth1 and decreased Prok2 expression in the ARDS patient (**Figure R15C**) compared to at-risk patients with pulmonary fibrosis (**Figure R15A**) as well as AECOPD (**Figure R15B**). We also revised the manuscript to add the data and agreed with the reviewer that this needs to be continuously investigated in the future study.

Figure R15. Fth1 and Prok2 expression in BALF neutrophils from different patients. (A-C) Representative fluorescence images of anti-Fth1/anti-Ly6G and anti-Prok2/anti-Ly6G-stained airway neutrophils from patients with pulmonary fibrosis (A), AECOPD (B) and ARDS (C). The nuclei were stained with DAPI, displayed in blue.

10. I could not read Suppl. Figures 7, 10 and 11; and Fig 3G, 4C, 8; and thus missed maybe some important information. They need to be presented in a more reader-friendly way.

Response: We apologize for not making it clear in the original manuscript. File conversion from Word to PDF may reduce image clarity. We have revised the figures, including the Supplementary Materials, to improve the clarity of the text contents.

Minor Comments

1. Results in Fig 3 need more detailed descriptions in the text.

Response: We appreciate the reviewer's helpful comment and we are truly sorry for not making it clear in the previous version. We have revised the results of Figure 3 and added more detailed descriptions accordingly: "...Lung tissues of WT and IL-10^{-/-} mice were obtained 24 h after LPS or PBS inhalation, rapidly digested to a single-cell suspension and analyzed following a single-tube protocol with unique transcript counting through barcoding with unique molecular identifiers (UMIs) using 10× Genomics Chromium platform (**Figure 3A**). After quality filtering, a total of 18,146 cells were analyzed for transcriptional profiling, for which an average of 1,268 genes per cell were measured. Among them, 1,129 cells from WT control mice, 4,478 cells from WT LPS-treated mice, and 6,518 cells from IL-10^{-/-} LPS-treated mice were profiled (**Figure 3B**). By using nonlinear dimensional reduction and graph-based clustering strategies, we visualized 18 subsets in two-dimensional t-distributed stochastic neighborhood embedding (t-SNE) projections (**Figure 3C**). We next sought out prototypical markers that defined cell populations and categorized these subsets into 9 major clusters, comprising epithelial cells (*Epcam*), endothelial cells (*Pecam1*), fibroblasts (*Col1a2*), neutrophils (*Ly6g*), monocyte-macrophages (*Fcgr1*, *Mrc1*), lymphocyte T-cells (*Cd3e*), lymphocyte B-cells (*Cd19*), dendritic cells (*Cd83*) and natural killer cells (*Klrb1c*) (**Figure 3D-F; Supplemental Figure 4**). These cell types demonstrated distinct top 15 enriched genes with typical corresponding biological

functions (**Figure 3G**). Notably, we observed dramatically increased proportion of neutrophils after LPS stimulation, and discrepant neutrophil subtypes could be identified between WT and IL-10^{-/-} ALI mice lungs.” (Page 8-9).

2. Methods: I think it is enough to mention the supplementary methods once and not state it for all assays.

Response: We thank the reviewer for the helpful comment and we have revised throughout the manuscript according to the suggestion.

3. Please provide a reference for the “reliable scoring system previously published” (Suppl. Method line 129).

Response: We appreciate the reviewer for pointing it out and we have provided the reference “Matute-Bello G, Downey G, Moore BB, et al. An official American Thoracic Society workshop report: features and measurements of experimental acute lung injury in animals. *Am J Respir Cell Mol Biol.* 2011;44(5):725-738. doi:10.1165/rcmb.2009-0210ST” and cited it in the revised Supplementary Materials for manuscript.

4. Statistics: Has normal distribution of data been tested? If so, what test has been used?

Response: We thank the reviewer for bringing up this helpful question. In our study, normal distribution of the samples was tested by Shapiro-Wilk test. We have added it in Statistics accordingly (Page 24).

5. Discussion: I think it is too early to speak of a new nomenclature system. The findings need to be reproduced and generalized first (valid also in other organs during other diseases etc etc) to introduce new systematic names for neutrophil subpopulations. In general, the new findings should be discussed in more detail, i.e. putting the data in context with the existing literature on known neutrophil subsets in different pathophysiological conditions. In this respect, the discussion remains somewhat superficial.

Response: We thank the reviewer's comment and find it very helpful, which inspires us to improve our manuscript. We are totally agreed that it is inappropriate to use 'a new nomenclature system' to describe what we found in the present study, so we decided to remove it in the revised version of Discussion.

We apologize for the lack of in-depth discussion in the previous version of the manuscript and we thank the reviewer for pointing it out. We have added more detailed discussion on the potential relations between known neutrophil subsets and our newly discovered ones according to the suggestion (page 20-21).

Reviewer 3:

Major Comments

1. One major concern is the lack of control group for PBS exposed IL-10 KO mice throughout the manuscript. This is necessary control group which need to be presented for these studies.

Response: We are truly sorry for not making it clear in the previous manuscript and appreciate the reviewer for pointing it out. We have revised **Figure 1** and **Figure 2** accordingly, which exhibited no significant difference of airway inflammation and lung injury between WT-PBS and IL-10^{-/-}-PBS control.

2. Statistical analyses are not well described and appear to be incorrectly chosen e.g., when looking for genotyping specific outcome with or without LPS exposure, then one way ANOVA is not suitable test for this purpose. Same comment stand for most of the data presented in this manuscript.

Response: We are truly sorry for our carelessness in the previous manuscript and appreciate the reviewer for pointing them out. We are totally agreed that one-way ANOVA is not suitable when looking for genotyping specific outcomes with or without LPS exposure since it contained two independent variables here, and we believe that two-way ANOVA with post hoc testing is more appropriate in this case. We have recalculated the data throughout the manuscript with adjusted statistical methods and revised them accordingly.

3. It appears authors performed scRNA seq on just 1-2 mice. If this is the case these analyses don't have enough rigor.

Response: We thank the reviewer for bring up the question. To address your issues, we repeated the scRNA-seq with the new method and the mice number in each group was

5. Briefly, lungs were carefully cut into small pieces and digested to get single-cell

suspensions, and tissue suspensions were passed through a 100- μ m cell strainer. Given that the present study primarily focused on immune cells, CD45⁺ immune cells were purified from lung tissue suspensions using CD45 microBeads kit (Miltenyi-Biotec) and enriched over MACS® Column. Freshly prepared CD45⁺ cell suspensions were performed immediately according to the manufacturer's protocol of 10 X Chromium 3' v3 kit (10x Genomics, Pleasanton, CA). Library was prepared and sequencing was performed on a NovaSeq 6000 platform (Illumina, Inc., San Diego, CA) in OE Biotech Co., Ltd. (Shanghai, China). Raw sequencing reads were transformed into fastq file with Illumina bcl2fastq2 Conversion Software v2.20 at <https://support.illumina.com/downloads/bcl2fastq-conversion-software-v2-20.html>, and quality checked with FastQC software v0.11.9, at <https://www.bioinformatics.babraham.ac.uk/projects/fastqc/>. Standard pipelines of cell ranger were used to do sequence processing, alignment to GRcm39 genome with default parameters (<https://support.10xgenomics.com/single-cell-gene-expression/software/pipelines/latest/>).

A total of 23,240 cells that passed quality control and filtering were analyzed for transcriptional profiling. Among them, 4,725 cells WT-PBS mice, 5,085 cells from IL-10^{-/-} KO-PBS mice, 6,263 cells from WT-LPS mice, and 7,167 cells from IL-10^{-/-} KO-LPS mice were profiled (**Figure R16A**). Then we categorized these leukocytes into 6 major clusters according to the prototypical markers, comprising neutrophils, monocyte-macrophages, lymphocyte T-cells, lymphocyte B-cells, dendritic cells (DC) and natural killer (NK) cells (**Figure R16B**). We also identified genes that specifically

characterized every major cell population (**Figure R16C**), therefore providing a landscape of ALI and IL-10-dependent cell subtypes based on single cell signatures.

With the nonlinear dimensional reduction and graph-based clustering strategies, we visualized 8 transcriptionally distinct clusters of neutrophils in two-dimensional t-distributed stochastic neighborhood embedding (t-SNE) projections (**Figure R17A**). Among them, N1-N4 mainly came from ALI mice, including N1, N3 primarily from WT ALI mice; and N2, N4 mainly from IL-10^{-/-} KO ALI mice (**Figure R17B**). To characterize the phenotypes of these subsets in detail, we performed single-cell differential expression analysis for each population and identified characteristic gene expression patterns for N1-N4 (**Figure R17C**). Based on their specifically enriched genes, we classified the neutrophils from ALI lungs into two populations: N1, N2 highly expressed *Fth1* and other genes (*Il12a*, *Adora2b*, etc) and were therefore referred to as Fth1^{hi} Neu; N3, N4 highly expressed *Prok2*, as well as *Scrg1* and *Serpinb1a*, and were defined as Prok2^{hi} Neu (**Figure R17D**). These data were in line with the results exhibited in Figure 3 and Figure 4A-G.

Fth1^{hi} Neu exhibited lower expression of *CD62L* and *Cxcr2*, and higher expression of *Cxcr4* and *Icam1* (**Figure R17E**), indicating a predominate ‘senescence’ phenotype. Besides, Fth1^{hi} cells exhibited higher expression in genes primarily associated with chemotaxis (*Ccl3*, *Ccl4*, *Fpr1*), pro-inflammatory cytokines (*Il1a*, *Il12a*, *Tnf*), inhibitors of apoptosis (*Bcl2a1a*, *Bcl2a1b*, *Bcl2a1d*), maintenance of iron homeostasis (*Fth1*, *Ftl1*, *Slc7a11*, *Slc3a2*) and anti-oxidative stress (*Sod2*, *Hmox1*, *Gstm1*). Comparatively, Prok2^{hi} Neu displayed a gene expression signature suggestive of

defense response to pathogens (*Ngp*, *Padi4*, *Serpinb1a*, *Prok2*, *S100a8*, *S100a9*), clearance of apoptotic cells (*Anxa1*, *Itgb2l*) and glycometabolism processes (*G6pdx*, *Ldha*, *Mpc1*, *Slc2a3*) (**Figure R17F**). This trend was essentially in agreement with Figure 5, thus verifying the accuracy and reliability of our scRNA-seq data in the manuscript.

EDITORIAL NOTE: FIGURES R16 AND R17 WERE REDACTED AT THE AUTHORS' REQUEST.

4. Human data is from a very small patient population and is presumably not well powered.

Response: We thank the reviewer for bring up the question. Recently, we collected the BALF of 11 inpatients from our department, among which 2 were pulmonary fibrosis, 2 were acute exacerbation of chronic obstructive pulmonary disease (AECOPD), 2 were pulmonary infection, and another 5 developed into ARDS within 48h after admission. For the BALF neutrophils, we also observed increased Fth1 and decreased Prok2 expression in the ARDS patient (**Figure R18C**) compared to at-risk patients with pulmonary fibrosis (**Figure R18A**) as well as AECOPD (**Figure R18B**). We also revised the manuscript to add the data and agreed with the reviewer that this needs to be continuously investigated in the future study.

Figure R18. Fth1 and Prok2 expression in BALF neutrophils from different patients. (A-C) Representative fluorescence images of anti-Fth1/anti-Ly6G and anti-Prok2/anti-Ly6G-stained airway neutrophils from patients with pulmonary fibrosis (A), AECOPD (B) and ARDS (C). The nuclei were stained with DAPI, displayed in blue.

5. Imaging data is presented as intensity per neutrophil which while helpful could bias the outcome when not considering the biological replicates/variability.

Response: We apologize for not making it clear in the original manuscript. In Figure 4, the mean fluorescence intensity of Fth1/Prok2 was calculated with more than 60 cells in each group from 6 mice. We have revised the figure legends accordingly. Additionally, in Figure 9, the neutrophils covered in statistics came from all the patients

mentioned in the manuscript, rather than the representative images. We have recalculated the mean fluorescence intensity because of the new collected inpatients and revised the related figure legends accordingly.

6. Survival data is not presented and need to be incorporated.

Response: We thank the reviewer for bring up the helpful question. According to your suggestion, we examined the effect of IL-10 on the survival rate using lethal doses of LPS (25 mg/kg). Survival status of the mice were recorded every day, which showed a lower survival rate in IL-10^{-/-} mice (26.7%) compared with WT mice (66.7%) (**Figure R19**). We also added the data in Supplementary Materials (**Supplemental Figure 3**) for the manuscript.

Figure R19. The effects of IL-10^{-/-} comparing to WT on survival of the severe ALI mice. Statistical analysis was performed using Log-rank test. N = 14-15 per group from 3 independent experiments. *p < 0.05 versus WT-LPS group.

7. IL-10 inhibition as utilized (concurrent with LPS) is neither a prophylactic nor a therapeutic strategy. In order to validate its roles as either its dosing regimen need to be adjusted.

Response: We appreciate the reviewer's valuable comments. Based on the research conducted by Kapur et al., recombinant mouse IL-10 was injected together with the antibodies as prophylactic strategy for transfusion-related acute lung injury²⁸. We have cited it in the revised manuscript. To further answer your question, recombinant murine IL-10 protein was administrated either 1 h prior to or after LPS challenge, followed by lung inflammation and tissue damage evaluation 24 h later. PBS was delivered in a similar fashion as control vehicle. As shown in **Figure R20A-C**, both pre- and post-administration decreased the total cell number, especially neutrophils, as well as total protein concentration and pro-inflammatory cytokine IL-6 in the BALF of ALI mice. In parallel, the lung injury in ALI mice was assessed by histological features from the pathological tissue slides of murine lungs. It could be observed that both pre- and post-administration alleviated LPS-induced tissue injury (**Figure R20D**). Furthermore, mice lung sections were stained with Fth1 and Ly6G, which indicated that Fth1 expression in ALI neutrophils might be reduced by IL-10 treatment either before or after LPS challenge (**Figure R20E**). Collectively, these data verified that both prophylactic (concurrent with or prior to LPS) and therapeutic (after LPS) IL-10 administration could prevent and rescue the development of LPS-mediated ALI in mice, which might be due to the lower expression of Fth1 in lung neutrophils.

Figure R20. Both pre- and post-administration of recombinant IL-10 protein alleviate inflammatory injury in LPS-induced ALI murine model. (A-C) IL-10 administration either prior to or after LPS challenge could reduce the total cell number, especially neutrophils (A), as well as IL-6 (B), and total protein concentration (C) in

the BALF of ALI mice. Data are presented as mean \pm SEM. Statistical analysis was performed using one-way ANOVA followed by Bonferroni's post hoc test. N = 5 per group from 3 independent experiments. *p < 0.05. **(D)** Histological images of the hematoxylin-eosin (H&E)-stained lung sections from distinct groups. Scale bar in the upper panel: 100 μ m; scale bar in the below panel: 40 μ m. **(E)** Representative images of anti-Fth1/anti-Ly6G immunofluorescence-stained lung tissues of ALI mice pre- or post-treated with recombinant IL-10. Nuclei were stained with DAPI, displayed in blue. Scale bar: 40 μ m.

Minor Comments

1. Methodology is presented for TEM evaluation of mitochondria, but images are not provided.

Response: We thank the reviewer for bring up the question. The TEM images of neutrophil mitochondria were shown in **Figure 6I** and we have replaced clearer images for your ease reviewing.

References

1. Yipp BG, *et al.* The Lung is a Host Defense Niche for Immediate Neutrophil-Mediated Vascular Protection. *Sci Immunol* **2**, (2017).
2. Morrissey SM, *et al.* A specific low-density neutrophil population correlates with hypercoagulation and disease severity in hospitalized COVID-19 patients. *JCI Insight* **6**, (2021).
3. Cabrera LE, *et al.* Characterization of low-density granulocytes in COVID-19. *PLoS Pathog* **17**, e1009721 (2021).

4. Lourda M, *et al.* High-dimensional profiling reveals phenotypic heterogeneity and disease-specific alterations of granulocytes in COVID-19. *Proc Natl Acad Sci U S A* **118**, (2021).
5. McLeish KR, *et al.* Differential Functional Responses of Neutrophil Subsets in Severe COVID-19 Patients. *Front Immunol* **13**, 879686 (2022).
6. Fiorentino DF, Bond MW, Mosmann TR. Two types of mouse T helper cell. IV. Th2 clones secrete a factor that inhibits cytokine production by Th1 clones. *J Exp Med* **170**, 2081-2095 (1989).
7. Saraiva M, Vieira P, O'Garra A. Biology and therapeutic potential of interleukin-10. *J Exp Med* **217**, (2020).
8. Jarry A, *et al.* Mucosal IL-10 and TGF-beta play crucial roles in preventing LPS-driven, IFN-gamma-mediated epithelial damage in human colon explants. *J Clin Invest* **118**, 1132-1142 (2008).
9. Ouyang W, O'Garra A. IL-10 Family Cytokines IL-10 and IL-22: from Basic Science to Clinical Translation. *Immunity* **50**, 871-891 (2019).
10. Lazear HM, Schoggins JW, Diamond MS. Shared and Distinct Functions of Type I and Type III Interferons. *Immunity* **50**, 907-923 (2019).
11. Blazek K, *et al.* IFN-lambda resolves inflammation via suppression of neutrophil infiltration and IL-1beta production. *J Exp Med* **212**, 845-853 (2015).
12. Luo Q, Liu Y, Liu S, Yin Y, Xu B, Cao J. Interleukin 28 is a potential therapeutic target for sepsis. *Clinical Immunology* **205**, 29-34 (2019).
13. Yuan L, Sun Y, Zhou N, Wu W, Zheng W, Wang Y. Dihydroquercetin Attenuates Silica-Induced Pulmonary Fibrosis by Inhibiting Ferroptosis Signaling Pathway. *Front Pharmacol* **13**, 845600 (2022).
14. Wang Y, *et al.* Hydrogen sulfide alleviates particulate matter-induced emphysema and airway inflammation by suppressing ferroptosis. *Free radical biology & medicine* **186**, 1-16 (2022).
15. Jiao Y, Yong C, Zhang R, Qi D, Wang D. Hepcidin Alleviates LPS-Induced ARDS by Regulating the Ferritin-Mediated Suppression of Ferroptosis. *Shock* **57**, 274-281 (2022).
16. Sasaki S, *et al.* Involvement of Prokineticin 2-expressing Neutrophil Infiltration in 5-Fluorouracil-induced Aggravation of Breast Cancer Metastasis to Lung. *Mol Cancer Ther* **17**, 1515-1525 (2018).

17. Yu X, *et al.* Identifying Prokineticin2 as a Novel Immunomodulatory Factor in Diagnosis and Treatment of Sepsis. *Crit Care Med* **50**, 674-684 (2022).
18. Darshan D, Vanoaica L, Richman L, Beermann F, Kuhn LC. Conditional deletion of ferritin H in mice induces loss of iron storage and liver damage. *Hepatology (Baltimore, Md)* **50**, 852-860 (2009).
19. Tang M, Chen Z, Wu D, Chen L. Ferritinophagy/ferroptosis: Iron-related newcomers in human diseases. *J Cell Physiol* **233**, 9179-9190 (2018).
20. Gao M, Monian P, Pan Q, Zhang W, Xiang J, Jiang X. Ferroptosis is an autophagic cell death process. *Cell Res* **26**, 1021-1032 (2016).
21. Negri L, Ferrara N. The Prokineticins: Neuromodulators and Mediators of Inflammation and Myeloid Cell-Dependent Angiogenesis. *Physiol Rev* **98**, 1055-1082 (2018).
22. Zhou QY. The prokineticins: a novel pair of regulatory peptides. *Mol Interv* **6**, 330-338 (2006).
23. Xu M, *et al.* c-MAF-dependent regulatory T cells mediate immunological tolerance to a gut pathobiont. *Nature* **554**, 373-377 (2018).
24. Huynh JP, *et al.* Bhlhe40 is an essential repressor of IL-10 during Mycobacterium tuberculosis infection. *J Exp Med* **215**, 1823-1838 (2018).
25. Sabat R, *et al.* Biology of interleukin-10. *Cytokine Growth Factor Rev* **21**, 331-344 (2010).
26. Shirakawa K, *et al.* IL (Interleukin)-10-STAT3-Galectin-3 Axis Is Essential for Osteopontin-Producing Reparative Macrophage Polarization After Myocardial Infarction. *Circulation* **138**, 2021-2035 (2018).
27. Kreisel D, *et al.* In vivo two-photon imaging reveals monocyte-dependent neutrophil extravasation during pulmonary inflammation. *Proc Natl Acad Sci U S A* **107**, 18073-18078 (2010).
28. Kapur R, *et al.* T regulatory cells and dendritic cells protect against transfusion-related acute lung injury via IL-10. *Blood* **129**, 2557-2569 (2017).

REVIEWERS' COMMENTS

Reviewer #1 (Remarks to the Author):

I am pleased to accept this work for publication. I thank the authors very much for their very complete answers and illustrated with many figures for each of my questions, which represent a lot of work.

Following this work and that corresponding to the other reviewer's requested, this work is very enriched!

Reviewer #2 (Remarks to the Author):

My concerns have been appropriately addressed. Great effort. Congratulations.

Reviewer #3 (Remarks to the Author):

I have no further comments.

Point-by point Responses to Reviewers' Comments:

Reviewer 1:

I am pleased to accept this work for publication. I thank the authors very much for their very complete answers and illustrated with many figures for each of my questions, which represent a lot of work.

Following this work and that corresponding to the other reviewer's requested, this work is very enriched!

Response: We greatly appreciate the reviewer's careful evaluation of our manuscript.

Reviewer 2:

My concerns have been appropriately addressed. Great effort. Congratulations.

Response: We greatly appreciate the comments from the reviewer.

Reviewer 3:

I have no further comments.

Response: We greatly appreciate the helpful suggestion from the reviewer.